# Modulation of sleep by trafficking of lipids through the *Drosophila* blood-brain barrier

**Fu Li[†], Gregory Artiushin[†], Amita Sehgal\***

Howard Hughes Medical Institute and Chronobiology and Sleep Institute, Perelman School of Medicine, University of Pennsylvania, Philadelphia, Perelman School of Medicine, University of Pennsylvania, Philadelphia, United States

**Abstract** Endocytosis through *Drosophila* glia is a significant determinant of sleep amount and occurs preferentially during sleep in glia of the blood-brain barrier (BBB). To identify metabolites whose trafficking is mediated by sleep-dependent endocytosis, we conducted metabolomic analysis of flies that have increased sleep due to a block in glial endocytosis. We report that acylcarnitines, fatty acids conjugated to carnitine to promote their transport, accumulate in heads of these animals. In parallel, to identify transporters and receptors whose loss contributes to the sleep phenotype caused by blocked endocytosis, we screened genes enriched in barrier glia for effects on sleep. We find that knockdown of lipid transporters *LRP1&2* or of carnitine transporters *ORCT1&2* increases sleep. In support of the idea that the block in endocytosis affects trafficking through specific transporters, knockdown of *LRP* or *ORCT* transporters also increases acylcarnitines in heads. We propose that lipid species, such as acylcarnitines, are trafficked through the BBB via sleep-dependent endocytosis, and their accumulation reflects an increased need for sleep.

**\*For correspondence:**
amita@pennmedicine.upenn.edu

[†]These authors contributed equally to this work

## Editor's evaluation

Through a convincing series of in vivo experiments, this work demonstrates that lipid- and carnitine transporters in glial cells of the *Drosophila* blood-brain barrier play important roles in the regulation of sleep. The data are consistent with metabolite clearance, in particular, the sleep-dependent transport of acyl-carnitine across the BBB, being an important function of sleep.

## Introduction

Although the regulation of sleep is normally studied on a behavioral and circuit level, there is increasing evidence for a role of basic cellular physiology. For instance, we found that disruption of endocytic and trafficking pathways in glia increases sleep in *Drosophila* (*Artiushin et al., 2018*). Glia of the *Drosophila* blood/hemolymph-brain barrier (BBB) emerged as a new cellular locus of sleep regulation in this study, such that genetically manipulating endocytosis in these cells alone was sufficient to increase sleep. As the increased sleep appeared to reflect higher sleep need, we asked if sleep enhances endocytosis through the BBB and found that this was indeed the case. However, the nature of the molecules trafficked through the BBB in a sleep-dependent manner was not known.

Barriers that separate the solutes of blood/hemolymph of the periphery from the interstitial fluid of the central nervous system display a rich profile of transporters, receptors, and trafficking proteins, which often reflect their unique functions. The *Drosophila* barrier glia populations share many conserved features with vertebrate barriers that employ endothelial and astrocytic populations (*DeSalvo et al., 2014*, *Weiler et al., 2017*). For instance, both are capable of moving lipids and

carbohydrates, ions, amino acids, and xenobiotics (*Weiler et al., 2017*). Furthermore, the fly barrier populations may serve specialized roles in metabolism, as not only the conduit of energy sources from the periphery, but also by containing the enzymatic machinery necessary for processing energy sources (*Volkenhoff et al., 2015*), and secreting signals in reference to nutritional state (*Chell and Brand, 2010*; *Spéder and Brand, 2014*).

Given that much of the traffic through the barrier involves energetic substrates, we conducted metabolomic profiling to identify candidate metabolites whose trafficking may be inhibited by an endocytosis block in glia. As genetic manipulations to block endocytosis can directly impact endocytosis-dependent carrier traffic as well as indirectly affect levels of membrane-associated transporters/receptors by altered recycling, we asked if specific glial proteins mediate effects of trafficking on sleep. Thus, we complemented the metabolomic approach with a knockdown screen to broadly search for barrier genes involved in sleep regulation. We report here that specific lipid and carnitine transporters act in barrier glia to affect sleep, and that disrupting expression of these transporters or of endocytosis leads to an accumulation of acylcarnitines in the head.

## Results

### Blocking endocytosis in glia induces acylcarnitine accumulation in fly heads

Our previous work showed that expression of *shibire (shi)*, a dominant negative dynamin that blocks endocytosis, in all or BBB glia increases sleep. To attain an unbiased, global assessment of metabolites that may be relevant to the increased sleep seen in *Repo-GAL4>20xShi.ts[1]* flies (hereafter referred to as *Repo>Shi[1]*), we conducted LC-MS analysis. Heads of male and female *Repo>Shi[1]* flies as well as *GAL4* and UAS controls were collected on dry ice with sieves and immediately frozen at –80°C. Each sample contained 200 fly heads (equal male and female), with five samples per genetic condition.

As an initial analysis, raw signal was scaled per each metabolite in reference to other samples within the dataset, and comparisons were made between Repo >Shi flies and each control, as well as controls to each other by Welch's t-test (*Supplementary file 1*).

**Table 1.** Fatty acid acylcarnitine accumulation in Repo>*Shi[1]* fly heads.
Acylcarnitine species in *Repo>Shi[1]* relative to parental controls. Welch's t-test was performed on the scaled signal for each metabolite, comparing the conditions shown.

| Sub pathway | Biochemical name | $\frac{Gal4>UAS}{Gal4Ctrl}$ | $\frac{Gal4>UAS}{UASCtrl}$ | $\frac{UASCtrl}{Gal4Ctrl}$ |
|---|---|---|---|---|
| | Acetylcarnitine (C2) | 4.25 | 2.68 | 1.59 |
| | Myristoylcarnitine (C14) | 28.81 | 26.39 | 1.09 |
| | Palmitoylcarnitine (C16) | 4.52 | 1.74 | 2.60 |
| | Palmitoleoycarnitine (C16:1) | 26.50 | 10.35 | 2.56 |
| | Margaroylcarnitine (C17) | 4.05 | 1.47 | 2.76 |
| | Stearoylcarnitine (C18) | 1.43 | 1.06 | 1.34 |
| | Linoleoylcarnitine (C18:2) | 6.77 | 3.14 | 2.15 |
| | Oleoylcarnitine (C18:1) | 4.00 | 1.92 | 2.08 |
| | Arachidoylcarnitine (C20) | 1.28 | 1.17 | 1.09 |
| | Behenoylcarnitine (C22) | 1.31 | 1.09 | 1.20 |
| | Eicosenoylcarnitine (C20:1) | 2.28 | 1.24 | 1.84 |
| | Lignoceroylcarnitine (C24) | 0.72 | 0.83 | 0.87 |
| | Nervonoylcarnitine (C24:1) | 2.36 | 1.26 | 1.87 |
| | Cerotoylcartinine (C26) | 0.85 | 0.97 | 0.87 |
| Fatty acid metabolism (acylcarnitine) | Ximenoylcarnitine (C26:1) | 0.75 | 0.88 | 0.85 |

Metabolites of interest were those for which signal from the experimental samples was significantly different, in the same direction, when compared to both controls, while controls compared to each were not significant. Of secondary interest were metabolites where a difference was seen in controls, but was proportionally smaller than consistent differences of each control to the experimental samples.

In surveying this dataset, the outstanding functional category, which contained multiple metabolites whose signal was consistently different in experimental animals versus controls, were the acylcarnitines (*Table 1*). Furthermore, the fold changes for given metabolites in this group, which consists of fatty acids conjugated to carnitine, were the highest overall. Carnitinylation occurs on fatty acids of various chain lengths, but only a subset of chain lengths in this dataset had sufficient signal, therefore we statistically compared *Repo>Shi[1]* flies to each of the parental controls for metabolites that had signal in at least three of five biological replicates for each genotype. Expression of *Shi[1]* in glia increased abundance in fly heads of the following acylcarnitine species: C2, C16, C16:1, C17, C18:1, C18:2 (*Figure 1A*). The only metabolite of this group with less signal in experimental animals was the longer chain, C24 (*Figure 1B*). Carnitine and deoxycarnitine were not significantly altered as compared to both controls.

To determine whether the blocking of endocytosis in glia affects acylcarnitine species specifically in heads or also in bodies, we measured the acylcarnitine abundance in bodies of *Repo>Shi[1]* and control flies. Bodies of male and female *Repo>Shi[1]* flies and parental controls were collected on dry ice with sieves and immediately frozen at –80°C. Each sample contained 50 fly bodies (male and female in equal numbers), with a total of five samples per genetic condition. Expression of *Shi[1]* in all glia increased abundance in fly bodies of the following acylcarnitine species: C4 butyryl, C4 isobutyryl, C4-OH butyryl, C4-OH isobutyryl, C12-OH, C14:1-OH, C14-OH,16:1-OH, C16-OH (*Figure 1—figure supplement 1*). These acylcarnitine species are different from those enriched in the head (*Figure 1A and B*); in particular, hydroxylated acylcarnitines are enriched in the body with a block in glial endocytosis. These data indicate that endocytosis block, which increases sleep need, also affects lipids in the body, but the nature of the lipids affected is different.

## Identification of barrier glia genes that affect sleep

In addition to identifying metabolites that accumulate as a result of blocked glial endocytosis, we sought to identify glial molecules whose function might be impacted by the block in endocytosis and thereby contribute to the effect on sleep. As noted above, glia of the BBB are the most relevant glial subtype for sleep-dependent endocytosis. We identified genes enriched in barrier glia by referring to transcriptional profiling that compared expression in the two glial populations that comprise the *Drosophila* BBB—subperineurial and perineurial glia (SPG+PG)— to all neurons, and all glia (*DeSalvo et al., 2014*). Preference was given to previously studied genes, particularly transporters, receptors, and those involved in trafficking, although many genes among the top 50 highly expressed in the barrier glia populations were also tested for effects on sleep. Of the genes enriched in barrier glia, we focused on those that showed low variability in expression from sample to sample. UAS-RNAi constructs for candidate genes were expressed with *Repo-GAL4 or RepoGS* drivers. Sleep in *Repo-GAL4* lines was compared with that in GAL4 and UAS alone controls, while sleep changes with *RepoGS* were determined by comparing flies maintained on RU486 with control flies tested without RU486. Knockdown of most genes did not produce a significant phenotype, but sleep was increased with knockdown of some transporter genes *CG3036, CG6126, mnd, VMAT, CG6836, Rh50, CG4462* (*Figure 2—figure supplement 1A*), cytoskeleton/trafficking factors *CG8036, Vha16, nuf* (*Figure 2—figure supplement 1B*), as well as *lsd-2* (*Figure 2—figure supplement 1A*), *acon*, and *MtnA* (*Figure 2—figure supplement 1D*). Meanwhile, knockdown of the transporter gene *CG16700* (*Figure 2—figure supplement 1A*), cytochrome P450 gene *Cyp6a20* (*Figure 2—figure supplement 1D*), and trafficking factor *Cln7* decreased total sleep (*Figure 2—figure supplement 1B*).

Since the candidate genes were selected based on enrichment within the barrier glia, we chose to examine and secondarily validate promising phenotypes through knockdown with more limited, barrier glia drivers. Thus, we screened a subset of the genes suggested by results of the pan-glial screen (*Figure 2—figure supplement 1*) with drivers *NP6293-GAL4* (*Awasaki et al., 2008*) and *Moody-GAL4* (*Strauss et al., 2015*) that target PG and SPG, respectively. Knockdown of *MtnA* (105011 KK), *CG6386* (108502 KK), *CG4462* (105566 KK), or *lsd-2* (102269 KK) did not significantly alter sleep when expressed in either of the barrier glial populations (data not shown). Reduction of

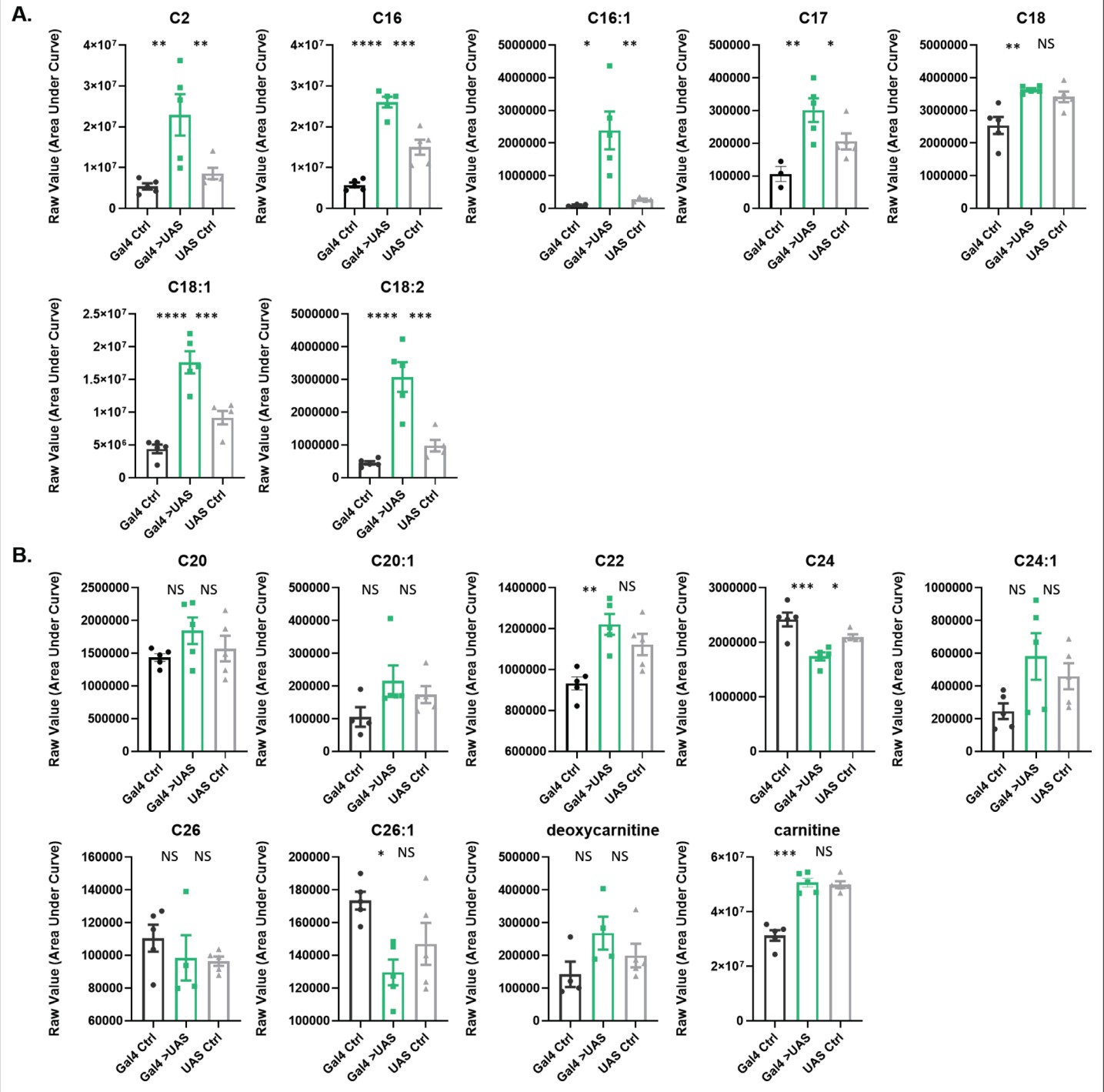

**Figure 1.** Acylcarnitine levels are increased in *Repo>Shi*[1] fly heads. (**A**) Short and long chain length or (**B**) very long chain length acylcarnitines from Repo>*Shi*[1] fly heads and parental controls. The raw signal from LC/MS is plotted, n=3–5 samples, of 200 fly heads each. One-way ANOVA, with Holm-Sidak post hoc comparisons. *p<0.05, **p<0.01, ***p<0.001, ****p<0.0001. Error bars represent standard error of the mean (SEM).

The online version of this article includes the following source data and figure supplement(s) for figure 1:

**Source data 1.** Acylcarnitine measurements in Repo>Shi1 and control fly heads.

**Figure supplement 1.** Levels of some acylcarnitines are increased in Repo>Shi[1] fly bodies.

Short, long chain and very long chain length acylcarnitines from Repo-GAL4>UAS-Shi[1] fly bodies and parental controls. The raw signal from LC-MS is plotted, n=5 samples of 50 fly bodies each. One-way ANOVA, with Holm-Sidak post hoc comparisons. *p<0.05, **p<0.01, ***p<0.001, ****p<0.0001. Error bars represent standard error of the mean (SEM).

**Figure supplement 1—source data 1.** Acylcarnitine measurements in Repo>Shi1 and control fly bodies.

*cyp6a20* in the PG population inconsistently reproduced the pan-glial sleep loss phenotype, so this was not pursued further (data not shown). The *VMAT* gene produces two isoforms, one of which is thought to be specific to glia (*Romero-Calderón et al., 2008*). Knockdown of *VMAT* (TRiP HMC02346) in the PG increased total sleep (*Figure 2A*), with sleep increases evident in both day- and nighttime (*Figure 2C and G*). However, locomotor activity during wake (activity index) was not different from that of controls (*Figure 2F*). Knockdown of *VMAT* in PG also did not significantly alter the average daytime or nighttime sleep bout length or bout number, although there was a trend toward increased bout length during the day (*Figure 2D, E, H, I*). At night, the increase seemed to result primarily from earlier onset of sleep after lights off.

In the SPG, *VMAT* knockdown had no effect (data not shown), which is consistent with protein expression, as the *VMAT-B* antibody specifically marks the PG (*DeSalvo et al., 2014*).

## Simultaneous knockdown of *Lrp1* and *Megalin* in barrier glia increases sleep

Although the candidate screen identified barrier genes whose knockdown increases sleep, as does blocking endocytosis in barrier glia, these genes were not obviously linked to the metabolite profile seen with a block in glial endocytosis. The results of metabolomic screening showed changes in lipid, and particularly acylcarnitine, trafficking. Therefore, we reassessed our screen candidates to consider transporters and receptors which may function in these pathways and could have been missed due to redundancy/lethality. *Lrp1* and *Megalin* (*Lrp2*) are two LDL receptor-related protein members involved in the transport of lipid carrier proteins at the fly barrier (*Brankatschk et al., 2014*). Expression is likewise found in mammals at the endothelial barrier (*Herz, 2003*).

Knocking down *Lrp1* and *Megalin* (*Lrp2*) individually in the pan-glial screen did not significantly alter total sleep time (*Figure 2—figure supplement 1A*). However, both *Lrp1* and *Megalin* (*Lrp2*) have been considered to be complementary (*Eraly and Nigam, 2002*; *Brankatschk et al., 2014*), therefore it is possible that inhibition of a single gene is insufficient to appreciably affect transport. Simultaneously knocking down *Lrp1* and *Megalin* in all glia with *Repo-GAL4* driver increased total sleep time, with increases in both day- and nighttime (*Figure 3A* and *Figure 3—figure supplement 1A and B*). Increases in sleep at night resulted from longer bout lengths, with a decrease in bout numbers (*Figure 3—figure supplement 1B*). Locomotor activity during wake (activity index) did not differ significantly from that of controls, indicating that sleep increase was not due to lethargy/sickness (*Figure 3—figure supplement 1C*). When both *Lrp* genes were knocked down with barrier glia drivers, a significant increase in total sleep was seen with the PG driver (*NP6293*) (*Figure 3B*), but not with either of the SPG drivers *moody* or *Rab9* (*Strauss et al., 2015*; *Artiushin et al., 2018*; *Figure 3C and D*). As with pan-glial knockdown, night sleep and bout length were increased and night sleep bout number was decreased when *Lrp* was knocked down with the *NP6293-GAL4* driver (*Figure 3—figure supplement 1B and E*). Meanwhile, the daytime sleep, bout length and bout number did not change (*Figure 3—figure supplement 1D*). Also, activity index of flies with *Lrp* knocked down in PG was not significantly different from that of controls (*Figure 3—figure supplement 1F*).

To target *Lrp1* and *Megalin* in barrier glia at the adult stage and thereby avoid developmental confounds, we coupled drivers specific to these glia with the temperature-sensitive *tubulin-Gal80* (*tub-Gal80^{ts}*) system that suppresses *GAL4* expression at 18°C but allows it at 31°C. In these experiments, to verify their role in sleep, we sought to use multiple RNAi lines for these transporters and were able to obtain three lines for *Megalin* (105071 KK, 36389 GD, and 27242 GD); however, for *Lrp1* only one line (8397 GD) had a chromosomal location that facilitated generation of double knockdowns. We combined each of the *Megalin* RNAi transgenes with the *Lrp* RNAi transgene and crossed them with the *9–137-GAL4* driver, which expresses in both PG and SPG (*DeSalvo et al., 2014*), and with *tub-Gal80^{ts}*. Knockdown of *Lrp1* (8397 GD) together with knockdown of *Megalin,* using either 105071 KK *or Megalin* 36389 KK, increased total sleep significantly in a temperature-dependent fashion, indicating an adult-specific role of these transporters in barrier glia (*Figure 3E, F, G and H* and *Figure 3—figure supplement 3D*). Surprisingly, in these flies, increases were during the day, and in one case (105071 KK) were accompanied by increased bout length and decreased bout number (*Figure 3—figure supplement 2D*). The activity index was not significantly different from that of controls (*Figure 3—figure supplements 2F; 3F*). However, *Lrp1* (8397 GD) with *Megalin* 27242 GD did not alter sleep (*Figure 3—figure supplement 4*), likely due to insufficient knockdown (see below).

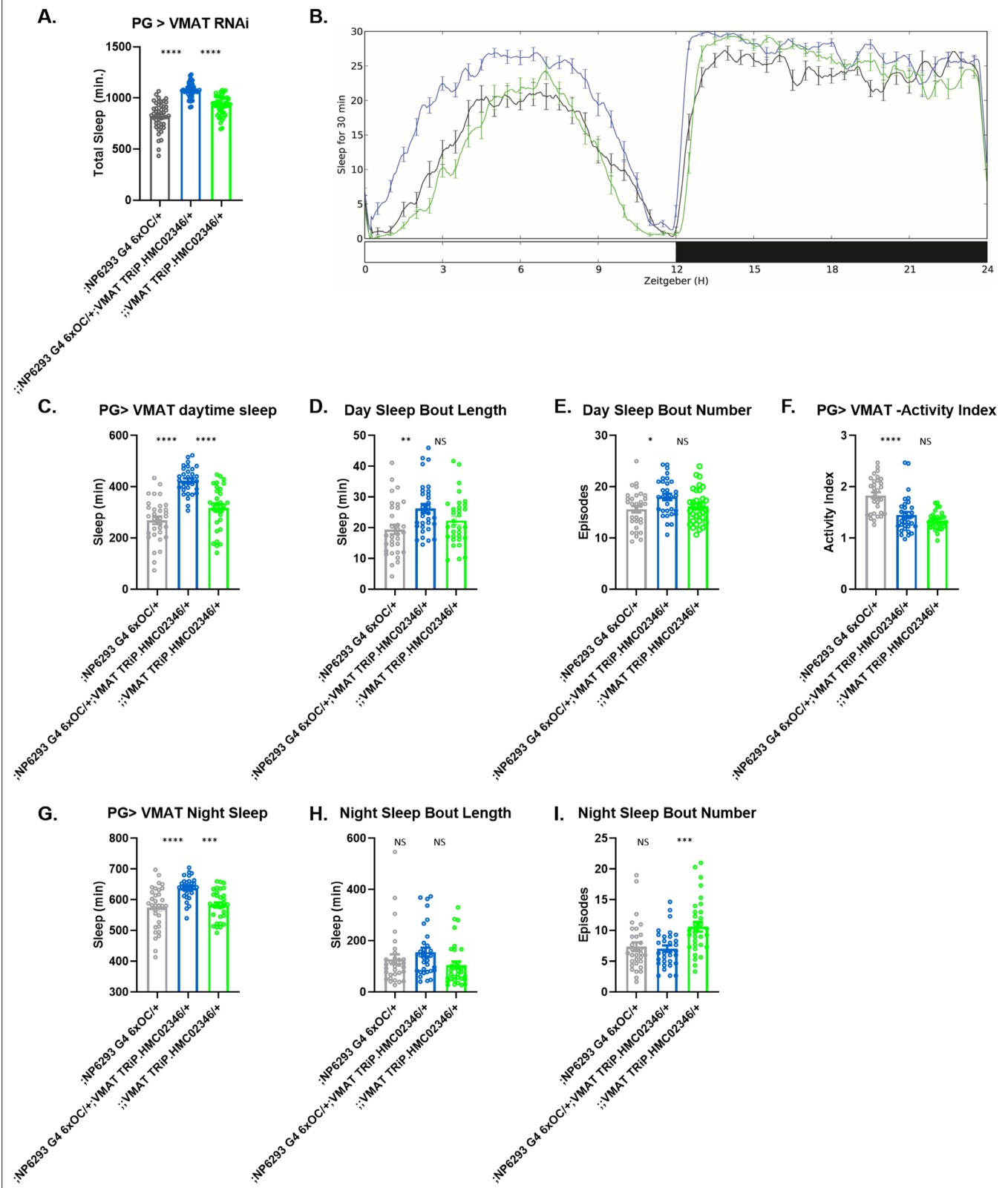

**Figure 2.** Change in sleep with knockdown of specific barrier-enriched genes in barrier glia. Total sleep (**A and B**) and sleep architecture (**C–I**) in female flies with knockdown of *VMAT*, UAS-HMC02346 driven by (perineurial glia [PG]) *NP6293-GAL4*. Mean sleep, sleep bout length and bout number are shown for daytime sleep (**C–E**) and nighttime sleep (**G–I**). Activity index, which depicts activity per waking minute, was not different from both controls

*Figure 2 continued on next page*

*Figure 2 continued*

(F). n=45–48 per genotype. One-way ANOVA, with Holm-Sidak post hoc comparisons. *p<0.05, **p<0.01, ***p<0.001, ****p<0.0001. Error bars represent standard error of the mean (SEM).

The online version of this article includes the following source data and figure supplement(s) for figure 2:

**Source data 1.** Sleep data of flies with VMAT knockdown in perineurial glia.

**Figure supplement 1.** Pan-glial RNAi screen of candidate genes enriched in the barrier glia.

**Figure supplement 1—source data 1.** Sleep data from the pan-glial RNAi screen of candidate genes enriched in the barrier glia.

As expected, most single knockdowns of *Lrp1* or *Megalin* did not alter sleep amount (*Figure 3—figure supplement 5A, B and E–J*; *Figure 3—figure supplement 6H*). However, *Lrp1* RNAi construct GD 109605 (*Figure 3—figure supplement 5C, D*) and *Megalin* RNAi construct GD 105387 (*Figure 3—figure supplement 6A, B*) individually increased total sleep when expressed by the driver *9–137-GAL4*. The GD 109605 construct was not effective at knocking down *Lrp1*, suggesting that sleep phenotypes are due to off-target effects (*Figure 3—figure supplement 7*). And the GD 105387 line affected sleep at the permissive temperature of 18°C, when the *GAL4* is not expected to be active, as well as at the restrictive temperature of 31°C, raising concerns about specificity.

Speculating that lack of a sleep phenotype with *Megalin* 27242 GD could be due to insufficient downregulation of *Megalin* gene expression, we sought to confirm the efficiency of knockdown of *Lrp1* and *Megalin* genes in the fly lines. As qPCR analysis is typically done with head RNA, and these genes are widely expressed, we tested the efficacy of the RNAi by expressing it with the ubiquitously expressed *Act-GAL4* driver. *Lrp1* RNAi fly lines 8397 GD, 13913 GD, and *Megalin* 36389 GD, 105387 KK, 105071 KK showed effective downregulation of their target genes (*Figure 3—figure supplement 7*). The *Megalin* gene in RNAi fly line 27242 GD was only slightly downregulated (*Figure 3—figure supplement 7B*), which is consistent with its lack of an effect on sleep (*Figure 3—figure supplement 4*).

To confirm that sleep alterations were specifically caused by the knockdown of *Lrp1* and *Megalin* in barrier glia, we asked if knockdown of *Lrp1* and *Megalin* in non-barrier glia had any effect on sleep. We used *MZ0709-GAL4*, *NP2222-GAL4*, and *Eaat-GAL4* drivers to express *Lrp1* and *Megalin* RNAi in ensheathing glia, cortex glia, and astrocyte-like glia, respectively, coupled with the *tub-Gal80^{ts}* system to restrict knockdown to adults. None of these manipulations affected sleep (*Figure 3—figure supplement 8*), showing that *Lrp1* and *Megalin* act only in barrier glia to regulate sleep.

## Knockdown of *Orct* and *Orct2* in barrier glia increases sleep

The organic cation (*Orct*) transporters are multi-substrate transporters whose substrates include carnitine (*Lahjouji et al., 2001*), and perhaps also carnitylated molecules, based on in vitro evidence for *Orct2* (*Kou et al., 2017*). However, like *Lrp1* and *Megalin* (*Lrp2*), *Orct* and *Orct2* are considered to be complementary (*Brankatschk et al., 2014*). As with the double knockdown of *Lrp1* and *Megalin*, knockdown of both *Orct* genes increased total sleep (*Figure 4A*). Simultaneous knockdown of *Orct2* and *Orct* in all glia with *Repo-GAL4* caused daytime and nighttime increase, the latter driven by increases in nighttime sleep bout length (*Figure 4—figure supplement 1A, B*). Locomotor activity during wake (activity index) did not change (*Figure 4—figure supplement 1C*). Testing their function in barrier glia showed that knockdown of *Orct1* and *-2* in either PG or SPG replicated the increased pan-glial sleep phenotype (*Figure 4B–D*), although this was only true with one line for the SPG (*Figure 4D*). Night-specific increases were noted with knockdown of *Orct2* 106681 KK and *Orct* 6782 GD in PG (*Figure 4—figure supplement 1E*).

Coupling each of two different *Orct* RNAis (6782 GD and 47133 GD) with *Orct2* 106681 KK RNAi and expressing them in adults using the *9–137-GAL4* driver with *tub-Gal80^{ts}* significantly increased sleep during the day and night (*Figure 4E, F, G and H*; *Figure 4—figure supplement 2* and *Figure 4—figure supplement 3D, E*). Day sleep bout length was increased in both pairs of knockdowns and night bout length also increased in *Orct2* 106681 KK coupled with *Orct* 47133 GD, accompanied by a decrease in sleep bout number (*Figure 4—figure supplements 2 and 3E*). However, *Orct2* 106681 KK coupled with *Orct* 6782 GD did not show changes in sleep bout length at night (*Figure 4—figure supplement 2*). As with other knockdown lines, activity index was not different from that of controls (*Figure 4—figure supplements 2 and 3*). Single knockdowns of *Orct* and *Orct2* genes did not change

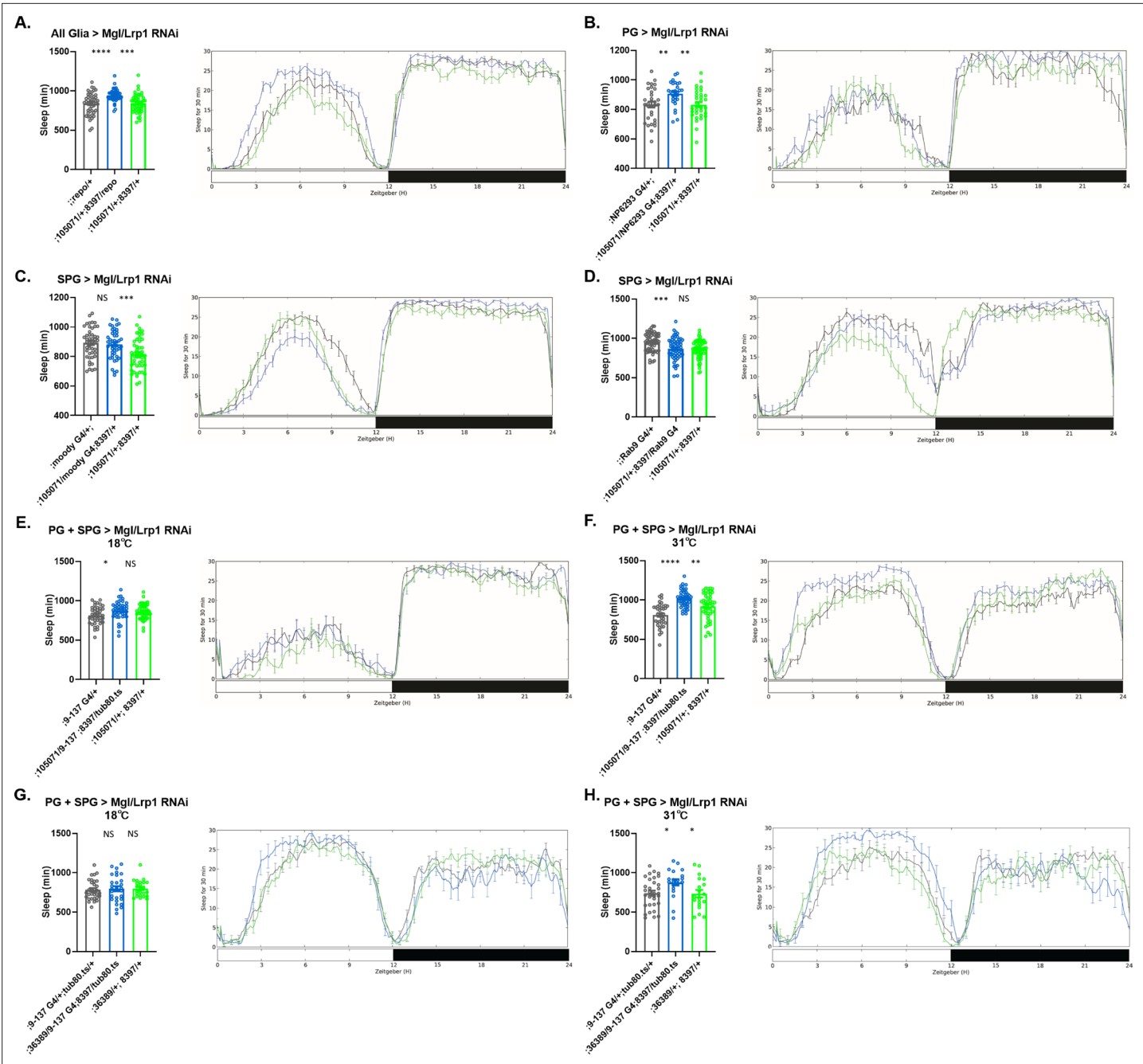

**Figure 3.** Sleep amount changes with knockdown of *Lrp1* and *Megalin* genes in all glia or barrier glia. Total sleep in female flies with knockdown of (**A**) *Lrp1* (8397 GD) and *Megalin* (105071 KK) RNAi driven by *Repo-GAL4*. n=45–48 per genotype; (**B**) *Lrp1* (8397 GD) and *Megalin* (105071 KK) RNAi driven by *NP6293-GAL4* (perineurial glia [PG]). n=45–48 per genotype; (**C**) *Lrp1* (8397 GD) and *Megalin* (105071 KK) RNAi driven by *moody-GAL4* (subperineurial glia [SPG]) n=45–48 per genotype; (**D**) *Lrp1* (8397 GD) and *Megalin* (105071 KK) RNAi expressed by *Rab9-GAL4* (SPG), n=45–48 per genotype; (**E**) *Lrp1* (8397 GD) and *Megalin* (105071 KK) RNAi driven by *9–137*-GAL4 (PG and SPG) with *tub-Gal80ᵗˢ* at the permissive temperature of 18°C, n=45–48 per genotype (**F**) *Lrp1* (8397 GD) and *Megalin* (105071 KK) RNAi driven by *9–137*-GAL4 (SPG and PG) with *tub-Gal80ᵗˢ* at the restrictive temperature of 31°C, n=45–48 per genotype. (**G**) *Lrp1* (8397 GD) and *Megalin* (36389 GD) RNAi driven by *9–137*-GAL4 (PG and SPG) with *tub-Gal80ᵗˢ* at the permissive temperature of 18°C, n=45–48 per genotype (**H**) *Lrp1* (8397 GD) and *Megalin* (36389 GD) RNAi driven by *9–137*-GAL4 (SPG and PG) with *tub-Gal80ᵗˢ* at the restrictive temperature of 31°C, n=45–48 per genotype. The same flies were assayed at permissive and restrictive temperatures. Comparisons were with one-way ANOVA, with Holm-Sidak post hoc. *p<0.05, **p<0.01, ***p<0.001, ****p<0.0001. Error bars represent standard error of the mean (SEM).

The online version of this article includes the following source data and figure supplement(s) for figure 3:

**Source data 1.** The sleep data of flies with simultaneous knockdown of Lrp1 and Megalin in all glia, perineurial glia , subperineurial glia or in perineurial

*Figure 3 continued on next page*

*Figure 3 continued*

glia plus subperineurial glia.

**Figure supplement 1.** Sleep/activity characteristics with knock down of *Lrp* in all glia or in perineurial glia (PG).

**Figure supplement 1—source data 1.** The sleep architecture data of flies with simultaneous knockdown of Lrp1 and Megalin in all glia and perineurial glia.

**Figure supplement 2.** Sleep/activity characteristics with knockdown of *Lrp* in perineurial glia (PG) and subperineurial glia (SPG).

**Figure supplement 2—source data 1.** The sleep architecture data of flies with simultaneous knockdown of Lrp1 and Megalin in perineurial glia and subperineurial glia.

**Figure supplement 3.** Sleep/activity characteristics with knockdown of *Lrp* in perineurial glia (PG) and subperineurial glia (SPG).

**Figure supplement 3—source data 1.** The sleep architecture data of flies with simultaneous knockdown of Lrp1(8397 GD) and Megalin (36389 GD) in barrier glia.

**Figure supplement 4.** Sleep amount changes with knockdown of *Lrp* genes in barrier glia.

**Figure supplement 4—source data 1.** The sleep data of flies with simultaneous knockdown of Lrp1(8397 GD) and Megalin (27242 GD) in barrier glia.

**Figure supplement 5.** Sleep amount is largely intact with knockdown of single *Lrp1* genes in surface glia.

**Figure supplement 5—source data 1.** The sleep data of flies with single knockdown of Lrp1 in barrier glia.

**Figure supplement 6.** Sleep amount is largely intact with knockdown of single Megalin genes in surface glia.

**Figure supplement 6—source data 1.** The sleep data of flies with single knockdown of Megalin in barrier glia.

**Figure supplement 7.** Effect of RNAi constructs on the expression of *Lrp1* and *Megalin* gene.

**Figure supplement 7—source data 1.** The expression of Lrp1 and Megalin genes in different RNAi fly lines carrying an Act-Gal4 driver.

**Figure supplement 8.** Sleep does not change with simultaneous knockdown of *Lrp1* and *Megalin* genes in ensheathing, astrocyte-like glia, or cortex glia.

**Figure supplement 8—source data 1.** The sleep data of flies with simultaneous knockdown of Lrp1 and Megalin in ensheathing, astrocyte-like glia or cortex glia.

sleep amount (*Figure 4—figure supplement 4A, B, E, F, G, H*), except for the *Orct* RNAi 52658 GD construct that increased sleep (*Figure 4—figure supplement 4C, D*) without changing expression of *Orct* (*Figure 4—figure supplement 6*), thereby rendering this result unreliable.

Checking the knockdown efficiency of *Orct* and *Orct2* RNAi constructs with the *Act-GAL4* driver revealed that *Orct* RNAi line 6782 GD and *Orct2* RNAi line 106681 KK significantly decreased target gene expression (*Figure 4—figure supplement 6A B*). Although simultaneous knockdown with *Orct* 47133 GD and *Orct2* 106681 KK increased sleep, we found that the 47133 GD line only slightly reduced *Orct* gene expression (*Figure 4—figure supplement 6A*).

We also used non-barrier-glia drivers *MZ0709-GAL4*, *NP2222-GAL4*, and *Eaat-GAL4* to express *Orct* and *Orct2* RNAi in ensheathing glia, cortex glia, and astrocyte-like glia, respectively. Simultaneously knocking down *Orct* and *Orct2* in ensheathing glia, cortex glia, and astrocyte-like glia did not change sleep time (*Figure 4—figure supplement 5*), demonstrating that only knockdown of *Orct* and *Orct2* in barrier glia increases sleep.

## Knockdown of *Lrp* and *Orct* genes in glia leads to accumulation of acylcarnitines

Given that knockdown of the *Lrp* and *Orct* genes in glia parallels effects of *Shi* expression in terms of increasing total sleep, we asked if it had the same effect on metabolite accumulation in fly heads. Based on the metabolomic results of *Repo >Shi[1,]* we chose to specifically assay acylcarnitines through LC-MS analysis. As in the case of the *Shi[1]* experiment, we collected heads from flies in which either *Lrp1* and *Megalin* or *Orct 1* and *Orct2* were knocked down with *Repo-GAL4*.

Knockdown of *Lrp1* and *Megalin* in all glia increased abundance in fly heads of the following acylcarnitine species: C3, C4 butyryl, C4-OH isobutyryl, C4 isobutyryl, C16, C18, C20 (*Figure 5*). Of these, C16 and a different form of C18 were also detected in *Repo>Shi[1]* flies. Longer-chain acylcarnitines, for example, those over C22, were largely undetected in the *Lrp* experimental samples, perhaps due to differences in methodological sensitivity (they were processed in a different facility from *Repo>Shi[1]* flies). Knockdown of *Orct* and *Orct2* similarly enriched acylcarnitines in fly heads. In particular, acylcarnitine species C16, C16:1, and C20 were increased significantly (*Figure 6*), with C16 and C16:1

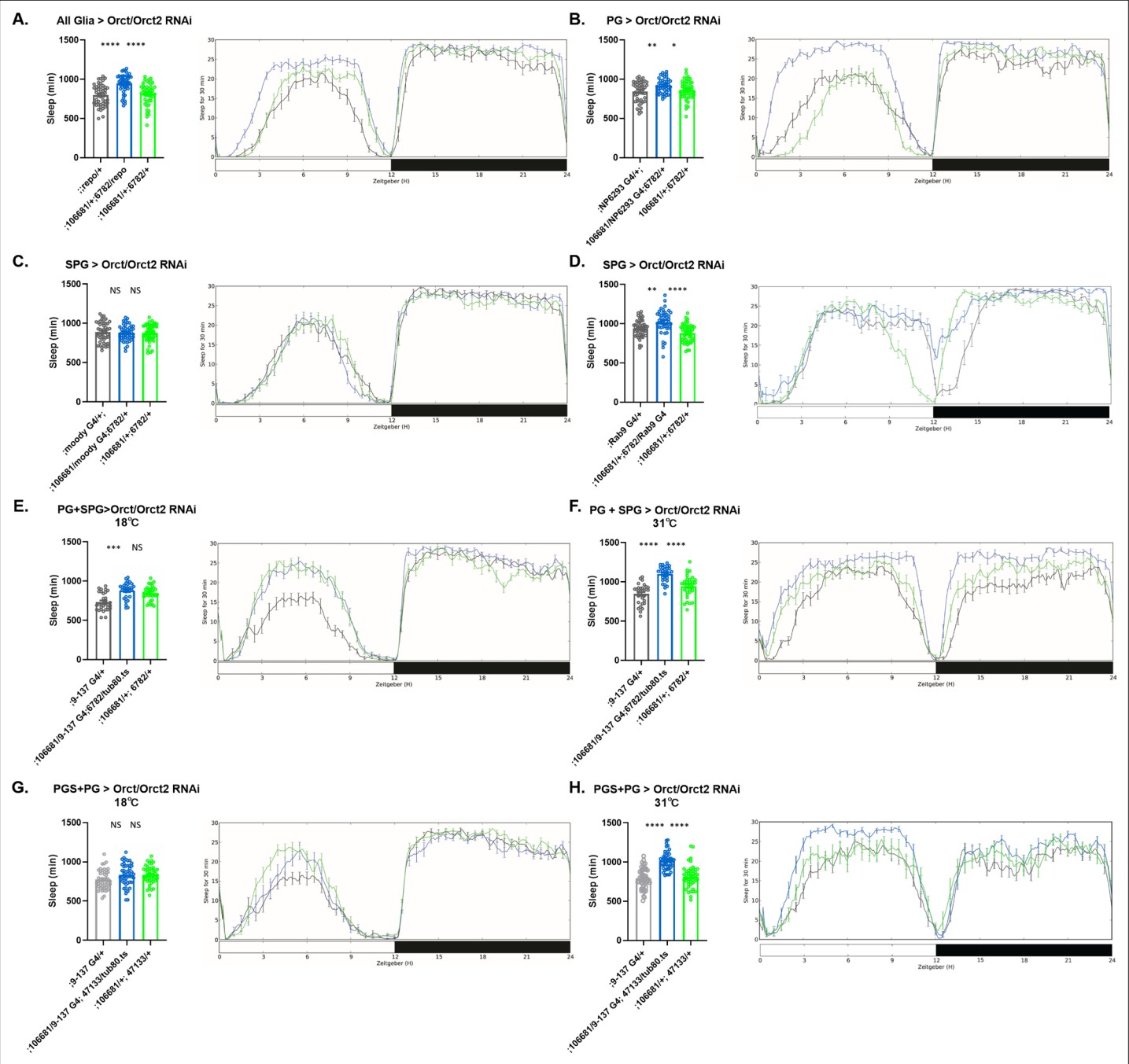

**Figure 4.** Sleep amount changes with knockdown of *Orct* genes in all glia or barrier glia. Total sleep in female flies with knockdown of (**A**) *Orct* (6782 GD) and *Orct2* (106681 KK) RNAi driven by *Repo-GAL4*. n=45–48 per genotype; (**B**) *Orct* (6782 GD) and *Orct2* (106681 KK) RNAi driven by *NP6293-GAL4* (perineurial glia [PG]). n=45–48 per genotype; (**C**) *Orct* (6782 GD) and *Orct2* (106681 KK) RNAi driven by *moody-GAL4* (subperineurial glia [SPG]) n=45–48 per genotype; (**D**) *Orct* (6782 GD) and *Orct2* (106681 KK) RNAi expressed by *Rab9-GAL4* (SPG), n=45–48 per genotype; (**E**) *Orct* (6782 GD) and *Orct2* (106681 KK) RNAi driven by *9–137*-GAL4 (PG and SPG) with *tub-Gal80$^{ts}$* at the permissive temperature of 18°C, n=45–48 per genotype; (**F**) *Orct* (6782 GD) and *Orct2* (106681 KK) RNAi driven by *9–137*-GAL4 (SPG and PG) with *tub-Gal80$^{ts}$* at the restrictive temperature of 31°C, n=45–48 per genotype; (**G**) *Orct* (47133 GD) and *Orct2* (106681 KK) RNAi driven by *9–137*-GAL4 (PG and SPG) with *tub-Gal80$^{ts}$* at the permissive temperature of 18°C, n=45–48 per genotype; (H) *Orct* (47133 GD) and *Orct2* (106681 KK) RNAi driven by *9–137*-GAL4 (SPG and PG) with *tub-Gal80$^{ts}$* at the restrictive temperature of 31°C, n=45–48 per genotype. Comparisons were with one-way ANOVA, followed by Holm-Sidak post hoc. *p<0.05, **p<0.01, ***p<0.001, ****p<0.0001. Error bars represent standard error of the mean (SEM).

The online version of this article includes the following source data and figure supplement(s) for figure 4:

**Source data 1.** The sleep data of flies that had simultaneous knock down of Orct and Orct2 in all glia or barrier glia.

*Figure 4 continued on next page*

*Figure 4 continued*

**Figure supplement 1.** Sleep/activity characteristics with knockdown of *Orct* in all glia or in perineurial glia (PG).

**Figure supplement 1—source data 1.** The sleep/activity architecture data of flies that had simultaneous knock down of Orct and Orct2 in all glia or perineurial glia.

**Figure supplement 2.** Sleep/activity characteristics with knock down of *Orct* in perineurial glia (PG) and subperineurial glia (SPG).

**Figure supplement 2—source data 1.** Sleep and sleep architecture data produced by knockdown Orct (6782 GD) and Orct2 (106681 KK) in barrier glia.

**Figure supplement 3.** Sleep/activity characteristics with knockdown of *Orct* in perineurial glia (PG) and subperineurial glia (SPG).

**Figure supplement 3—source data 1.** Sleep and sleep architecture data produced by knockdown of Orct (47133 GD) and Orct2 (106681 KK) in barrier glia.

**Figure supplement 4.** Sleep amount is largely intact with knockdown of single *Orct* genes in surface glia.

**Figure supplement 4—source data 1.** Sleep data of flies with single knockdown of Orct or Orct2 in barrier glia.

**Figure supplement 5.** Sleep does not change with simultaneous knockdown of *Orct* and *Orct2* genes in ensheathing, astrocyte-like glia, or cortex glia.

**Figure supplement 5—source data 1.** Sleep data of flies with simultaneous knock down of Orct and Orct2 in ensheathing, astrocyte-like glia or cortex glia.

**Figure supplement 6.** Expression of *Orct* and *Orct2* genes in RNAi fly lines.

**Figure supplement 6—source data 1.** The expression of Orct and Orct2 RNA in fly lines where Orct or Orct2 RNAi were driven by Act-Gal4 driver.

matching results of *Repo>Shi[1]* and C16 also overlapping with *Repo>Lrp*. Similar to the case of the *Lrp* samples, longer-chain acylcarnitines over 22 were not detected well. Overall, while the total sets of species affected are not identical, blocking transporters of lipids or carnitines yields similar increases in acylcarnitines to those seen with blocking endocytosis in glia, with certain species being affected in all manipulations.

## Acylcarnitine feeding has a small effect on sleep

The accumulation of acylcarnitines in the brains of *Repo>Shi[1]* flies or flies with decreased expression of *Lrp1* and *Megalin* or *Orct* and *Orct2* in barrier glia raised the possibility that acylcarnitines contribute to the sleep phenotypes of these flies; in other words that they increase sleep or sleep need. To investigate whether supplementing fly food with acylcarnitines could alter sleep, we fed flies each of three mixes of acylcarnitines of varying chain lengths, Mix 1 (A-141, Sigma), 2 (A-144, Sigma), and 3 (A-145, Sigma). Acylcarnitine Mix 1 contains 10 mg/mL of C0 and C2 in methanol, of which C2 was increased in heads of Repo>Shi[1] flies and of short-sleeping mutants *fmn*, *rye*, and *sssP1* (*Bedont et al., 2023*). Acylcarnitines in Mix 2 (C4, iC4, C5, iC5, C6, C8, C10, C12, C14, C18) and Mix 3 (C8:1, C10:1, C12:1, C14:1, C14:2, C16:1, C18:1, C18:2) are at 100–200 µg/mL concentration in 90:10 methanol: 1% aqueous formic acid. Mix 2 contains iC4 and C18 that were found increased with the *Lrps* pan-glial knockdown and in *fmn*, *rye*, and *sssP1*, while Mix 3 has the C16:1, C18:1, and C18:2 species found in Repo>Shi[1], Repo>Orcts and in sleep short mutants *fmn*, *rye*, and *sssP1*. No commercial products are available for the very long chain acylcarnitines, such as C20, C22, C24, C24:1, and C26:1, that increase with pan-glial knockdown of *Lrps* or *Orcts* or in *fmn*, *rye*, and *sssP1*. Each of the acylcarnitine mixes was added to the fly food (5% sugar and 1.8% agar) of experimental iso31 flies and similar concentrations of methanol or methanol and aqueous formic acid were added to the food of controls. Acylcarnitine Mix 1 was tested at five different concentrations of 2.5 mg/L, 5 mg/L, 10 mg/L, 25 mg/L, and 50 mg/L. As the LC-MS analysis indicated that the concentration of C2 is approximately 100–1000 times higher than that of the medium or long chain acylcarnitines, acylcarnitine Mix 2 and 3 were tested at concentrations of 0.025 mg/L, 0.05 mg/L, 0.10 mg/L, 0.25 mg/L, and 0.5 mg/L (*Figure 7*). All experimental and control flies were monitored for sleep for 3 days after the feeding.

Despite a slight increase in sleep at the 2.5 mg/L concentration, significant changes in sleep were not seen with any concentration of Mix 1 (*Figure 7A*, *Figure 7—figure supplement 1*). Low concentrations of acylcarnitine Mix 2 0.025 mg/L and 0.05 mg/L increased daytime sleep amount slightly (*Figure 7B*, *Figure 7—figure supplement 2A, E*) and significantly increased nighttime sleep (*Figure 7—figure supplement 2B, F*), in case of 0.05 mg/L with a decrease in nighttime sleep bout number, suggesting that sleep is more consolidated. The 0.1 mg/L concentration of acylcarnitine Mix 2 increased daytime sleep and sleep bout length, again suggesting more consolidation (*Figure 7—figure supplement 2*). Low concentrations of acylcarnitine Mix 3—0.025 mg/L, 0.05 mg/L, and 0.1 mg/L—also increased

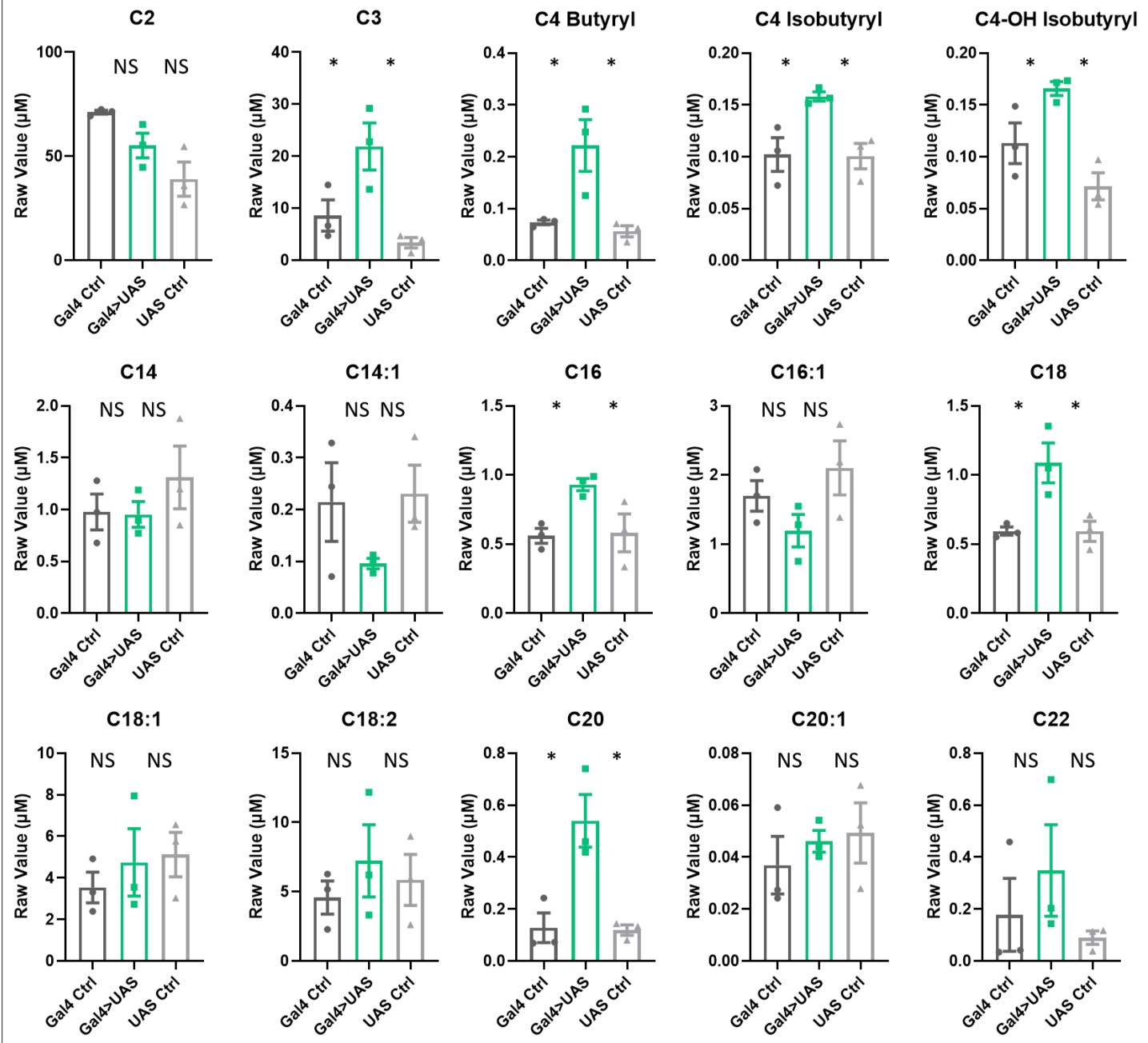

**Figure 5.** Acylcarnitine levels are increased in Repo>Lrp*1*+*Mgl* RNAi fly heads. Short, medium, and long chain length acylcarnitines from *Repo-GAL4>Lrp1+Mgl* RNAi fly heads and parental controls. The raw signal from LC/MS is plotted, n=3 samples, of 300 fly heads each. One-way ANOVA, with Holm-Sidak post hoc comparisons. *p<0.05, **p<0.01. Error bars represent standard error of the mean (SEM).

The online version of this article includes the following source data for figure 5:

**Source data 1.** Acylcarnitine measurements in fly heads with simultaneous knockdown of Lrp1 and Megalin in all glia.

daytime sleep with increased daytime sleep bout length (*Figure 7—figure supplement 4A, D, E, H, I and L*). The 0.1 mg/L concentration of acylcarnitine Mix 3 increased nighttime sleep as well (*Figure 7—figure supplement 4J*). Activity index decreased in flies fed low concentrations of Mix 2 or Mix 3 alone, raising the possibility that lower mobility contributes to the increased sleep; however, results of experiments where flies were fed Mix 2 and 3 together alleviated this concern (see below). High concentrations of acylcarnitine Mix 2 or 3 had no effect on sleep (*Figure 7C*).

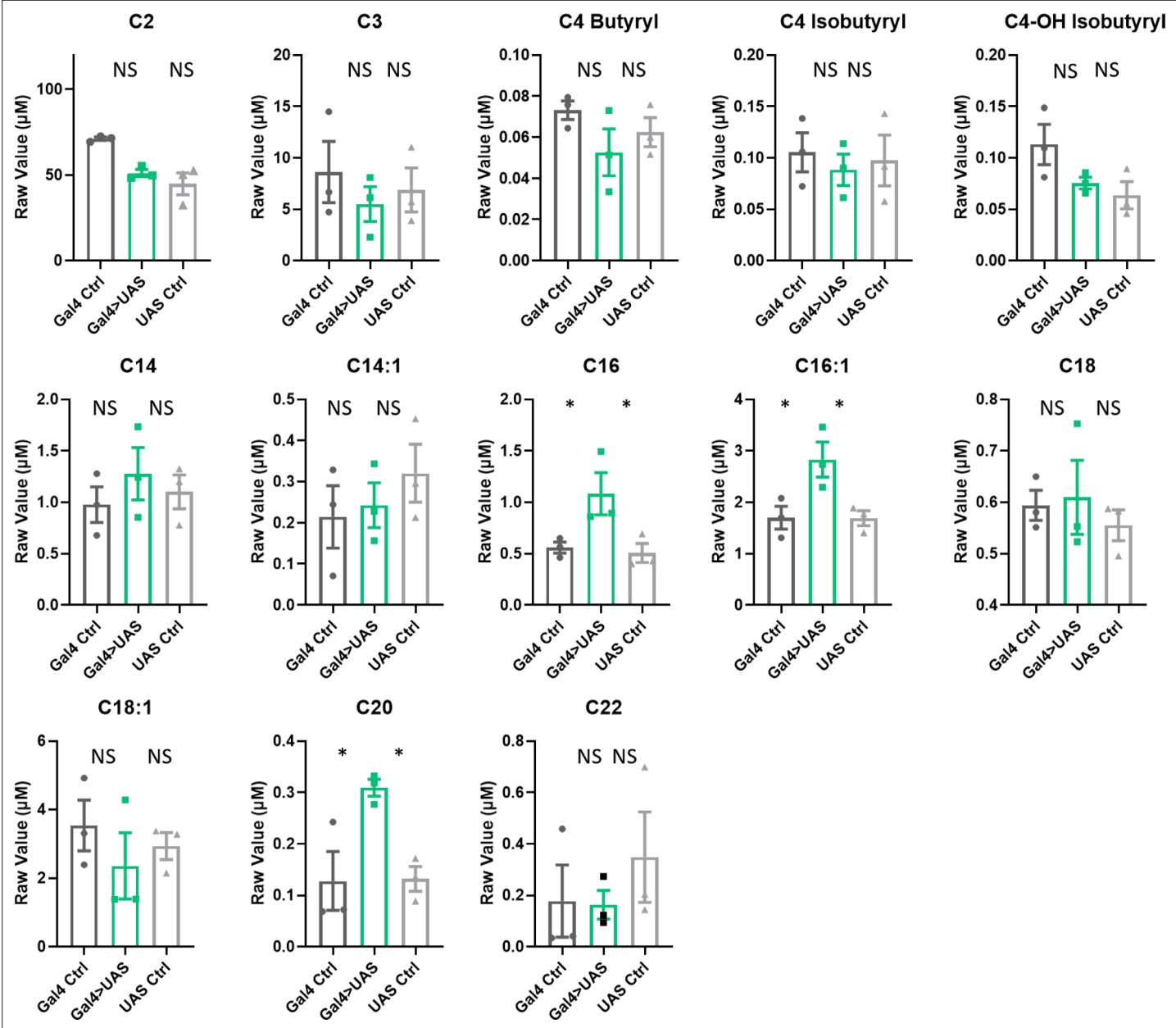

**Figure 6.** Acylcarnitine levels are increased in Repo>*Orct2+Orct* RNAi fly heads. Short, medium, and long chain length acylcarnitines from *Repo-GAL4>Orct + Orct2* RNAi fly heads and parental controls. The raw signal from LC/MS is plotted, n=3 samples, of 300 fly heads each. One-way ANOVA, with Holm-Sidak post hoc comparisons. *p<0.05. Error bars represent standard error of the mean (SEM).

The online version of this article includes the following source data for figure 6:

**Source data 1.** Acylcarnitine measurements in fly heads with simultaneous knockdown of Orct and Orct2 in all glia.

Since each of the long chain acylcarnitine mixes, Mix 2 or Mix 3, increased sleep, we asked if the two together had more robust effects on fly sleep. Combinations of Mix 2 and Mix 3 were tested at concentrations of 0.0125 mg/L, 0.025 mg/L, 0.05 mg/L, 0.10 mg/L, 0.25 mg/L, and 0.5 mg/L (*Figure 8*). As with each mix alone, low concentrations of Mix 2 and 3 together increased sleep amounts (*Figure 8A and B*), but high concentrations had no effect (*Figure 8D and F*) except for a slight increase with 0.25 mg/L (*Figure 8E*).

The magnitude of the change in sleep was not noticeably different, as compared to the individual mixes, suggesting that sleep-promoting components in the two mixes are redundant rather than additive. As in the case of the feeding of individual mixes, we also examined sleep architecture—daytime

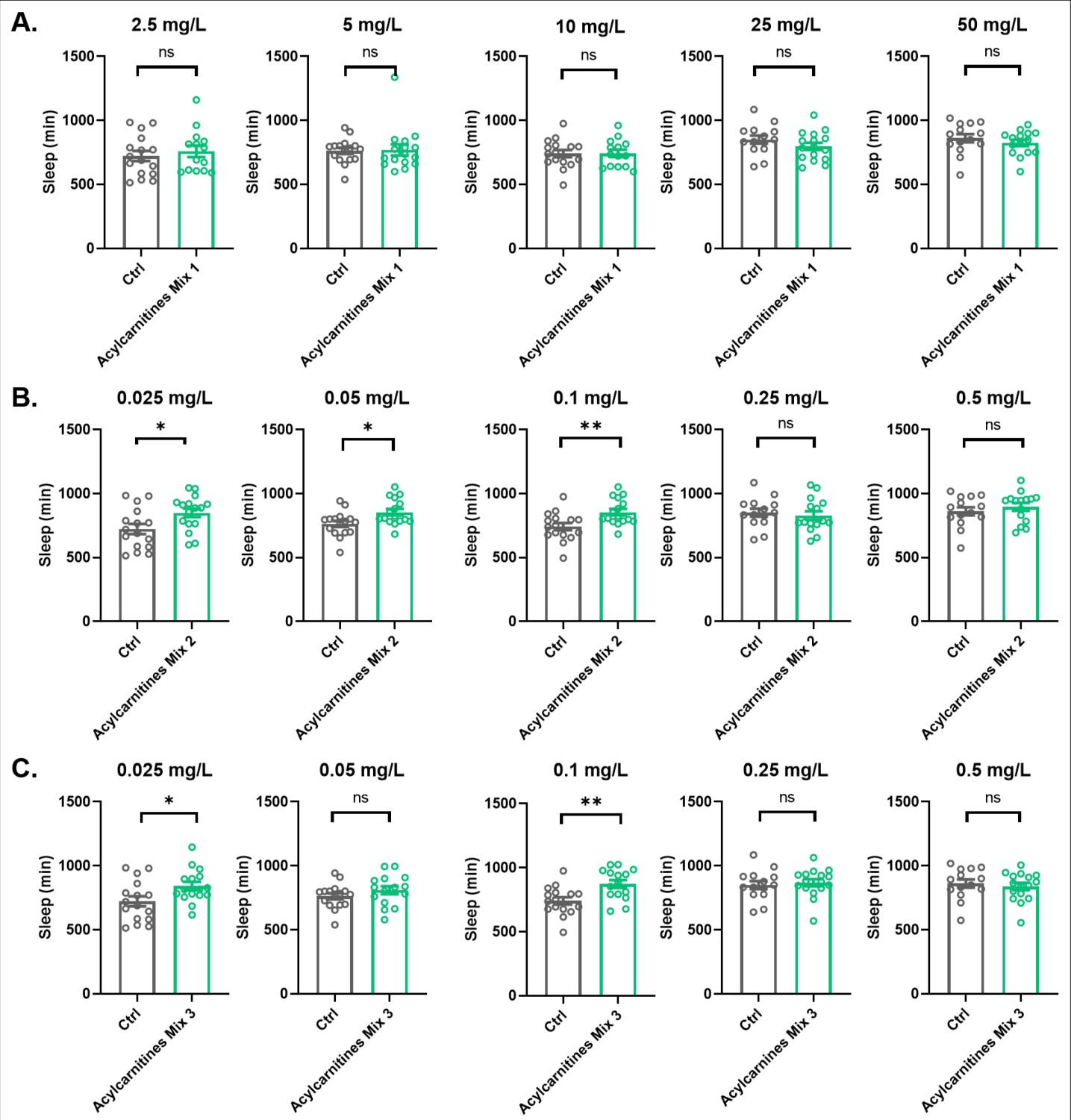

**Figure 7.** Effects of acylcarnitine Mix 1, 2, or 3 on sleep. Total sleep was evaluated in female iso31 flies fed with mixes of short or long chain length acylcarnitines or control food. (**A**) Acylcarnitine Mix 1 at 2.5 mg/L, 5 mg/L, 10 mg/L, 25 mg/L, and 50 mg/L concentrations. (**B**) Acylcarnitine Mix 2 at 0.025 mg/L, 0.05 mg/L, 0.10 mg/L, 0.25 mg/L, and 0.5 mg/L concentrations. (**C**) Acylcarnitine Mix 3 at 0.025 mg/L, 0.05 mg/L, 0.10 mg/L, 0.25 mg/L, and 0.5 mg/L concentrations. N=16 per condition. Mann-Whitney T test was performed. *$p<0.05$, **$p<0.01$. Error bars represent standard error of the mean (SEM).

The online version of this article includes the following source data and figure supplement(s) for figure 7:

**Source data 1.** Sleep data of flies fed food containing acylcarnitines Mix 1, 2 or 3 or control solvent.

*Figure 7 continued on next page*

*Figure 7 continued*

**Figure supplement 1.** Sleep amount with acylcarnitine Mix 1 feeding.

**Figure supplement 1—source data 1.** Sleep data of flies fed food containing acylcarnitine Mix 1 or control solvent.

**Figure supplement 2.** Sleep/activity characteristics with feeding of low concentrations of acylcarnitine Mix 2.

**Figure supplement 2—source data 1.** Sleep data of flies fed food containing low concentration of acylcarnitine Mix 2 or control solvent.

**Figure supplement 3.** Sleep/activity characteristics with feeding of high concentrations of acylcarnitine Mix 2.

**Figure supplement 3—source data 1.** Sleep data of flies fed food containing high concentration of acylcarnitine Mix 2 or control solvent.

**Figure supplement 4.** Sleep/activity characteristics with feeding of low concentrations of acylcarnitine Mix 3.

**Figure supplement 4—source data 1.** Sleep data of flies fed with food containing low concentration of acylcarnitine Mix 3 or control solvent.

**Figure supplement 5.** Sleep/activity characteristics with feeding of high concentration of acylcarnitine Mix 3.

**Figure supplement 5—source data 1.** Sleep data of flies fed with food containing high concentration of acylcarnitine Mix 3 or control solvent.

and nighttime sleep bout length and number and activity index (activity per waking minute)—in flies fed Mix 2 and 3 together (*Figure 8—figure supplements 1 and 2*). In general, changes in day or night sleep were accompanied by increased sleep bout length and decreased bout number, indicating better sleep consolidation. Flies fed Mix 2 and 3 together showed either no change or even an increase in activity index, demonstrating an effect on sleep independent of overall reduced mobility.

## Discussion

Our previous work suggested that endocytosis through the BBB is a function of sleep, but the nature of the molecules trafficked remained unknown. Using a dual-pronged approach of a targeted genetic screen and unbiased metabolomic analysis, we report here that the passage of lipids through the BBB is important for sleep. Blocking such transport increases sleep in conjunction with an accumulation of acylcarnitines.

Through a pan-glial RNAi knockdown screen of candidate genes expressed in the fly barrier, we identified molecules that affect daily sleep amount. In initial follow-up experiments, we targeted knockdown to each barrier layer separately, which is subject to the concern that behavioral phenotypes requiring simultaneous knockdown in both barrier populations would be missed. Nevertheless, we consider this scenario to be less probable, as permeability through the populations is quite different, with smaller solutes likely passing through the PG but not the tight barrier of the SPG, lipophilic solutes, or xenobiotics potentially passing through each uninhibited, and larger solute requiring endocytic mechanisms that likely have to work in each population in tandem. Therefore, in most cases of knockdown, transport would either be inhibited by the one barrier population essential for it or would be interrupted by either population. This was indeed the case for the *VMAT* isolated in the initial screen. The lipid and carnitine transporters we subsequently focused on also produced phenotypes through knockdown in a single barrier layer, although we also tested effects of knockdown in both layers. While there was redundancy at the level of the transporters, the two barrier layers did not compensate for each other.

An additional consideration of preliminarily screening with pan-glial drivers is that if knockdown in multiple glial subtypes has opposing effects on sleep, we may have obscured a role for the barrier cells. Again, we attempted to minimize this risk by primarily selecting genes whose expression is both highly abundant and specifically enriched in the barrier populations, as opposed to the transcriptome dataset of all glial cells (*DeSalvo et al., 2014*). The assumption is that multifold expression in the barrier populations is indicative of prevailing importance in these cells, although this is a caveat.

The vesicular monoamine transporter (*VMAT*) has previously been identified as a target of reserpine, which promotes sleep in the fly (*Nall and Sehgal, 2013*). *VMAT* mutants exhibit higher baseline sleep, and also lose less sleep than controls when subject to mechanical sleep deprivation. *VMAT* can traffic multiple monoamines such as dopamine, serotonin, histamine, and octopamine, but no single neuronal population or neurotransmitter system was implicated as responsible for the *VMAT* sleep phenotype (*Nall and Sehgal, 2013*). In flies, *VMAT* exists as two isoforms, *VMAT-A*, which is expressed in monoaminergic neurons, and *VMAT-B*, which appears to be specific to PG (*DeSalvo et al., 2014*), as it is also found in fenestrated glia in the visual system (*Romero-Calderón et al., 2008*), which are a

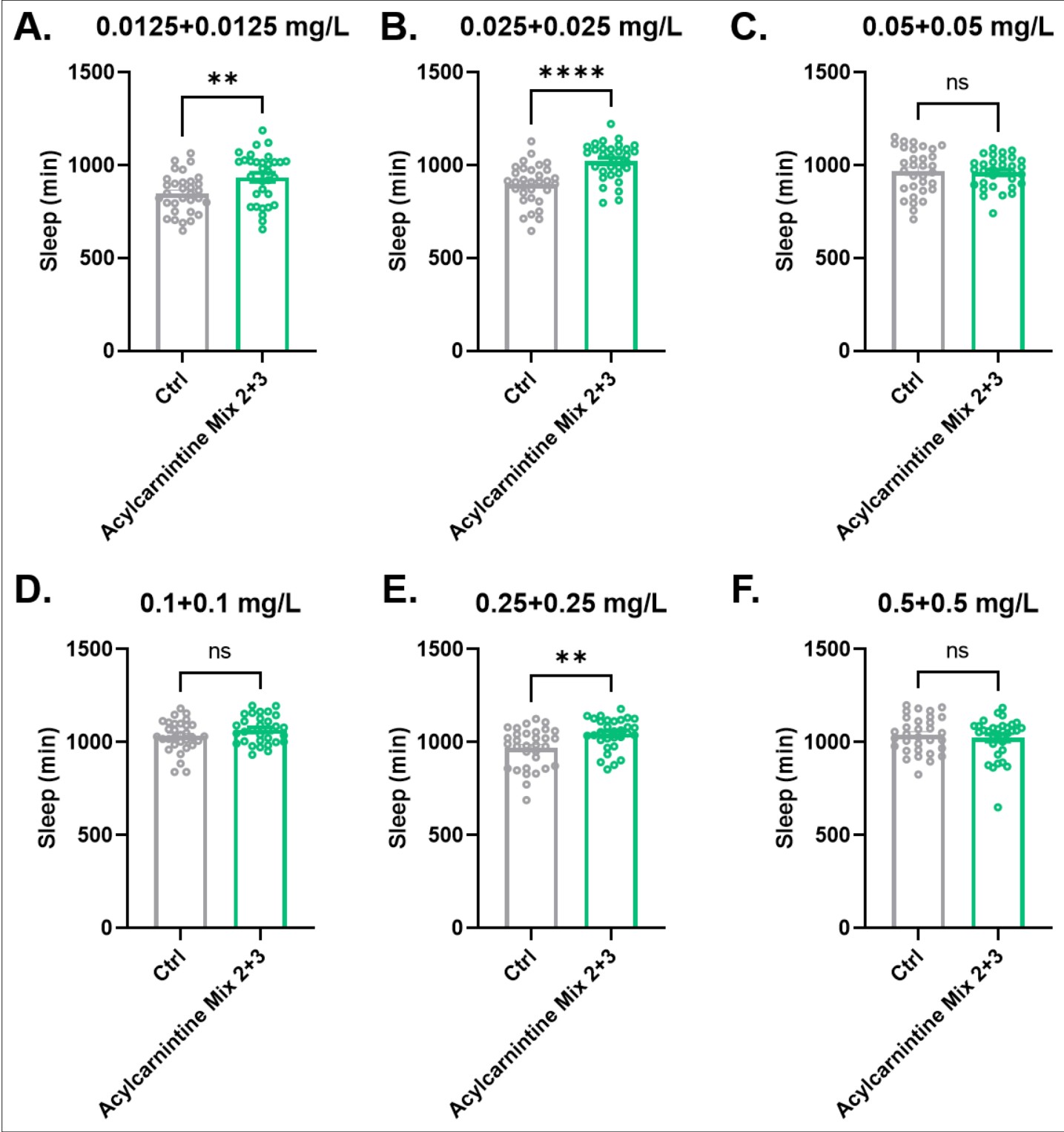

**Figure 8.** Effects of acylcarnitine Mix 2 and 3 on sleep. Total sleep was evaluated in female iso31flies fed with mixes of long chain length acylcarnitines or control food. Each of acylcarnitine Mix 2 and 3 was added at a concentration of 0.0125 mg/L (**A**), 0.025 mg/L (**B**), 0.05 mg/L (**C**), 0.10 mg/L (**D**), 0.25 mg/L (**E**), or 0.5 mg/L (**F**). n=32 per condition. Mann-Whitney T test was performed. *p<0.05, **p<0.01, ****p<0.0001. Error bars represent standard error of the mean (SEM).

The online version of this article includes the following source data and figure supplement(s) for figure 8:

**Source data 1.** Sleep data of flies fed with food containing acylcarnitines Mix 2 and 3 or control solvent.

*Figure 8 continued on next page*

*Figure 8 continued*

**Figure supplement 1.** Sleep/activity characteristics with feeding of low concentrations of acylcarnitine Mix 2 and Mix 3.

**Figure supplement 1—source data 1.** The sleep/activity architecture of flies fed with food containing low concentrations of acylcarnitines Mix 2 and 3 or control solvent.

**Figure supplement 2.** Sleep/activity characteristics with feeding of high concentrations of acylcarnitine Mix 2 and Mix 3.

**Figure supplement 2—source data 1.** The sleep/activity architecture of flies fed with food containing high concentrations of acylcarnitines Mix 2 and 3 or control solvent.

specialized form of PG (*Kremer et al., 2017*). It is unknown whether *VMAT-B* would function similarly in glia as *VMAT-A* does in neurons. *VMAT-B* contains an additional cytoplasmic domain, which has been suggested to promote retention in the plasma membrane as opposed to trafficking to vesicles (*Greer et al., 2005*). *VMAT* knockdown increased sleep, as did disrupting endocytic trafficking at the barrier (*Artiushin et al., 2018*). Whole-brain levels of monoamines were not altered in flies that expressed *Shi*[1] in glia, nevertheless it is possible that this gross analysis would not be sensitive to local changes at the barrier. In the visual system glia, *VMAT-B* may be necessary for uptake of histamine (*Romero-Calderón et al., 2008*). Interestingly, histamine is known to alter permeability of the BBB in mammals (*Lu et al., 2010*).

Our metabolomic data indicated that acylcarnitines are elevated in the heads of the long-sleeping *Repo>Shi*[1] flies. Acylcarnitines are transported to mitochondria for fatty acid oxidation, but are also secreted as they are found in plasma in mammals (*Schooneman et al., 2013*). Given that circulating acylcarnitines can be taken up by cells, we investigated *Lrp1/Megalin* and *Orct/Orct2* as candidate transporters for this uptake. *Lrp1* and *Megalin* are lipoprotein carrier receptors known to function in the fly barrier (*Brankatschk et al., 2014*) and although knockdown of each one separately in all glia did not affect sleep, reducing expression of both in all glia or barrier glia increased sleep. Likewise, knockdown of *Orct* and *Orct2,* which are homologs of the human carnitine transporters (*Eraly and Nigam, 2002*) and transport carnitine as well as acylcarnitines (*Pochini et al., 2004*), increases sleep. LC-MS analysis shows that knockdown of these transporters enriches acylcarnitines in fly heads just as blocking endocytosis does, supporting the idea that *LRP* and *ORCT* are among the proteins affected by the block in endocytosis. Exactly where the accumulation of acylcarnitines occurs is not known at this time, but we suggest that it is largely extracellular. In addition, given that heads contain brains as well as fat body, we acknowledge that the fat body may contribute to changes in acylcarnitines and other metabolites reported here. It is further possible that the differences in lipid species elevated in head as opposed to body samples may reflect the different outputs of the head and abdominal fat bodies, respectively.

Acylcarnitine accumulation in flies with blocked glial endocytosis or lipid transport could occur as a consequence of the high sleep in these animals. We believe this is unlikely as an increase in acylcarnitines appears to generally occur under conditions of sleep deprivation, that is conditions that would promote sleep. Carnitine conjugation of long chain fatty acids was reported in cortical metabolites of sleep-deprived mice, while short and medium chain fatty acids were reduced (*Hinard et al., 2012*). Changes in acylcarnitines were also noted in the peripheral blood of sleep-deprived or sleep-restricted humans (*Davies et al., 2014*; *Weljie et al., 2015*) as well as over a day:night cycle (*Ang et al., 2012*; *Dallmann et al., 2012*). We find too that a common feature of short-sleeping fly mutants, which are models for chronic sleep deprivation, is an increase in acylcarnitines, regardless of the mechanism that causes the sleep loss (*Bedont et al., 2023*). In the metabolomic analysis reported here, samples were collected at ZT 0–3, and so not after a period of sleep deprivation; however, all the manipulations (knockdown of *Orct/Lrps* or expression of *Shi*) increased sleep need, and so may be associated with high acylcarnitines at all times. Thus, acylcarnitines are generally associated with sleepiness rather than sleep. While this does not necessarily mean that they promote sleep, we show here that feeding flies low concentrations of medium and long chain acylcarnitines modestly increases sleep, although the fate of these acylcarnitines or where they act to affect sleep is not known. High concentrations have no effect, either due to some feedback mechanism or perhaps because high levels of the methanol solvent inhibit feeding. Regardless of their role in sleep regulation, we suggest that acylcarnitines are a key marker of sleep need across species, and could be exploited for this

purpose. Sleep is likely required to reduce levels of acylcarnitines, perhaps as part of a broader role in metabolic homeostasis.

## Materials and methods
### Fly stocks

The initial screen was performed with lab stock drivers:;;*Repo-GAL4*/TM6c, *Sb* and UAS-*Dicer*; *Repo*GeneSwitch. SPG driver *Moody-GAL4* and surface driver *9–137-GAL4* were shared by Roland Bainton, while PG driver *NP6293-GAL4* was a gift of Marc Freeman.;;UAS-*20xShi.ts1* (referred to as UAS-*20xShi[1]*) was shared by Gerald Rubin. RNAi lines were ordered from VDRC (KK and GD collection) and Bloomington (TRiP collection) stock centers, with the stock number provided in the below table.

| Iso31 | Laboratory Stocks | |
|---|---|---|
| w; 9–137-GAL4; tub-Gal80$^{ts}$ | Laboratory Stocks | |
| w; MZ0708-GAL4; tub-Gal80$^{ts}$ | Laboratory Stocks | |
| w; Eaat1-GAL4; tub-Gal80$^{ts}$ | Laboratory Stocks | |
| w; tub-Gal80ts; NP222-GAL4 | Laboratory Stocks | |
| y[1] w[*]; PBac{y[+mDint2] w[+mC]=I-SceI(FRT.Rab9-GAL4.ATG(loxP.3xP3-RFP))}VK00033/TM3, Sb[1] | Bloomington Stock Center | 51587 |
| w;; Repo-GAL4, 6x crossed to Iso | Bloomington Stock Center | 7415 |
| y[1] v[1]; P{y[+t7.7] v[+t1.8]=TRiP.HMC02346}attP2 | Bloomington Stock Center | 44471 |
| y[1] sc[*] v[1] sev[21]; P{y[+t7.7] v[+t1.8]=TRiP.HMC03067}attP2 | Bloomington Stock Center | 50666 |
| y[1] v[1]; P{y[+t7.7] v[+t1.8]=TRiP.JF01350}attP2 | Bloomington Stock Center | 31381 |
| y[1] v[1]; P{y[+t7.7] v[+t1.8]=TRiP.JF01628}attP2 | Bloomington Stock Center | 31151 |
| y[1] sc[*] v[1] sev[21]; P{y[+t7.7] v[+t1.8]=TRiP.HMC05119}attP40 | Bloomington Stock Center | 60125 |
| y[1] sc[*] v[1] sev[21]; P{y[+t7.7] v[+t1.8]=TRiP.HMC04670}attP40 | Bloomington Stock Center | 57583 |
| Orct GD | VDRC | 52658 |
| Orct2 GD | VDRC | 1176 |
| Orct GD | VDRC | 47133 |
| Orct2 KK | VDRC | 106681 |
| Orct GD | VDRC | 6782 |
| Megalin KK | VDRC | 105071 |
| Lrp1 GD | VDRC | 8397 |
| Lrp1 KK | VDRC | 109605 |
| Lrp1 GD | VDRC | 13913 |
| LRP1 GD | VDRC | 3710 |
| Megalin GD | VDRC | 27242 |
| Megalin GD | VDRC | 105387 |
| Megalin GD | VDRC | 36389 |
| CG3036 | VDRC | 108500 |
| CG3168 | VDRC | 48011 |
| CG4797 | VDRC | 10598 |
| CG3897 | VDRC | 101083 |
| L(2)80717 GD | VDRC | 11117 |

*Continued on next page*

*Continued*

| Iso31 | Laboratory Stocks | |
|---|---|---|
| CG6126 GD | VDRC | 7326 |
| Lsd-2 KK | VDRC | 102269 |
| CG16700 KK | VDRC | 110058 |
| Mct1 KK | VDRC | 106773 |
| Jhl-21 KK | VDRC | 108509 |
| Mnd KK | VDRC | 110217 |
| Lrp1 KK | VDRC | 106364 |
| CG5687 GD | VDRC | 33262 |
| CG6386 KK | VDRC | 108502 |
| CG4462 KK | VDRC | 105566 |
| Jhl-21 KK | VDRC | 108509 |
| Mnd KK | VDRC | 110217 |
| Lrp1 KK | VDRC | 106364 |
| CG5687 GD | VDRC | 33262 |
| CG6386 KK | VDRC | 108502 |
| CG4462 KK | VDRC | 105566 |
| CG4462 GD | VDRC | 6786 |
| Npc2b GD | VDRC | 16652 |
| Catsup KK | VDRC | 100095 |
| CG8036 KK | VDRC | 105633 |
| Arc1 KK | VDRC | 109141 |
| Vinc KK | VDRC | 105956 |
| Ctp KK | VDRC | 109084 |
| Vha16 KK | VDRC | 104490 |
| Blos2 KK | VDRC | 107449 |
| Capu KK | VDRC | 110404 |
| Snapin GD | VDRC | 28145 |
| Nuf KK | VDRC | 104172 |
| Moe KK | VDRC | 110654 |
| Cln7 kk | VDRC | 109291 |
| Ggamma30A KK | VDRC | 102706 |
| Nplp3 KK | VDRC | 105584 |
| InR GD | VDRC | 992 |
| Sdr KK | VDRC | 105549 |
| Grip KK | VDRC | 103551 |
| Olf86-m GD | VDRC | 17657 |
| Wgn GD | VDRC | 9152 |
| sesB KK | VDRC | 104576 |

*Continued*

| Iso31 | Laboratory Stocks | |
|---|---|---|
| Acon GD | VDRC | 11767 |
| Fer1HCH KK | VDRC | 102406 |
| Fer2LCH KK | VDRC | 106960 |
| MtnA KK | VDRC | 105011 |
| Tctp GD | VDRC | 45532 |
| Neu3 KK | VDRC | 102041 |
| Ppn KK | VDRC | 108005 |
| Stall KK | VDRC | 104737 |
| CG10433 GD | VDRC | 41162 |
| Arouser KK | VDRC | 105755 |
| Gste7 KK | VDRC | 104977 |
| DnaJ-1 KK | VDRC | 104618 |
| Argk GD | VDRC | 34037 |
| Santa-maria GD | VDRC | 33153 |
| 14-4-3 E KK | VDRC | 108129 |
| Cyp6a20 GD | VDRC | 3313 |
| Pld KK | VDRC | 106137 |
| CG10126 KK | VDRC | 44104 |
| CG10184 KK | VDRC | 104488 |
| CG7800 KK | VDRC | 100221 |
| AdamTS-A GD | VDRC | 33347 |
| Hnf-4 GD | VDRC | 12692 |
| CG9743 KK | VDRC | 108185 |
| CG7461 KK | VDRC | 110220 |

## Behavior

Flies were crossed and raised on standard cornmeal-molasses medium in bottles under 12:12 hr light:dark. Offspring were entrained at 25°C, in LD12:12 conditions until at least 6 days post-eclosion, before age-matched flies, which were group-housed in bottles, were used in sleep assays. Mated females were loaded into glass locomotor tubes with 2% agar 5% sugar. Sleep was quantified by the *Drosophila* Activity Monitor (DAM) system, using the established definition of a minimum 5 min of inactivity. Data was analyzed in PySolo (*Gilestro and Cirelli, 2009*).

For the sleep experiments using *9–137*-GAL4, *MZ0709-GAL4, NP2222-GAL4,* or *Eaat-GAL4* coupled with *tub-Gal80ts*, the same group flies were monitored initially at 18°C for 3 days and then shifted to 31°C for 2 days and shifted back to 18°C again.

## Metabolomics

Approximately 7-day-old Repo>*Shi¹ flies* were maintained at 1°C for 2 days to impair endocytosis in all glia. At ZT 0–3, whole flies were quickly frozen in Falcon tubes chilled on dry ice, and placed at –80°C. Each tube contained 50 flies. Heads were subsequently removed from the body by briefly vortexing the tube. Heads were then separated from the rest of the body by an array of copper sieves, whose housing was buried in dry ice to keep the preparation cool. For each sample, 200 fly heads, of equal parts from males and females, were collected in 1.5 mL tubes which were quickly refrozen. Samples were shipped on dry ice to Metabolon, Inc, where they were assessed by LC-MS (*Evans et al., 2009*).

For metabolomic analysis of *Lrp* and *Orct* knockdown fly lines, each sample contained 300 fly heads, of equal parts from males and females. Samples were processed by LC-MS at the Penn Metabolomics Core.

For the comparison of acylcarnitine levels in the bodies, 50 bodies of equal numbers from males and females were collected and quickly frozen in Eppendorf tubes chilled on dry ice, and placed at –80°C. Samples were processed by LC-MS at the Penn Metabolomics Core.

## qPCR validation of *Lrps* or *Orcts* RNAis

To determine the knockdown efficiency of target transcripts with *Lrp* and *Orct* RNAis, the *Lrp* and *Orct* RNAi flies were crossed with *Act-GAL4* driver to express RNAi ubiquitously. RNA was extracted from experimental and control fly lines by using the SV total RNA extraction kit and its protocol (Z3105, Promega). The RNAs were resuspended in nuclease-free water about 300 ng/μL. cDNA libraries of RNAi fly lines and control iso31 were synthesized by using the High Capacity cDNA Reverse Transcription Kit and its protocol (4368814, Applied Biosystems). cDNAs from *Lrps* and *Orcts* RNAi and iso31 were amplified using SYBR Green PCR mix (4364346, Lifetech) on an Applied Biosystems ViiA7 qPCR machine. Primers are listed below. We calculated relative transcript levels by ddCT.

| *Lrp1* forward | **TGCTCCCTGCTCTTCAGTTG** |
|---|---|
| *Lrp1* reverse | CTCAGTGAAAAATTGCTGCGG |
| *Megalin* forward | CGCCATTCACAGCATCACAG |
| *Megalin* reverse | AGGTTGCTGATGGCCACAAG |
| *Orct* forward | ACTTCGGGAGTCTTCCTGGTT |
| *Orct* reverse | GCCGTGAGCATAAAGCCCA |
| *Orct2* forward | GAGTGGAACCTGGTTTGTGG |
| *Orct2* reverse | CCGTACTTATCGCTCAGTTGAC |

## Acylcarnitine mixture feeding

Three kinds of acylcarnitine mixtures Mix 1, Mix 2, and Mix 3 were ordered from Sigma (A-141, A-144, and A-145, Sigma). These acylcarnitine mixtures, which used methanol or methanol plus formic acid as solvents, were added individually into the fly food (5% sugar and 1.8% agar) and the same concentrations of methanol or methanol and aqueous formic acid were added into control food. Iso31 flies were fed five concentrations of acylcarnitine Mix 1—2.5 mg/L, 5 mg/L, 10 mg/L, 25 mg/L, and 50 mg/L. Mix 2 or Mix 3 was added into food at concentrations of 0.025 mg/L, 0.05 mg/L, 0.10 mg/L, 0.25 mg/L, and 0.5 mg/L, respectively. For feeding of acylcarnitine mixtures of Mix 2 and Mix 3, each of Mix 2 and Mix 3 was added at a concentration of 0.0125 mg/L, 0.025 mg/L, 0.05 mg/L, 0.10 mg/L, 0.25 mg/L, and 0.5 mg/L, as done for separate feeding of Mix 2 or Mix 3. For the sleep assay, ~7-day-old iso31 flies were housed individually in glass tubes containing acylcarnitine or methanol or methanol and aqueous formic acid control food in Percival incubators. As above, sleep was determined based upon activity recorded with the Trikinetics DAM system (http://www.trikinetics.com/). All flies were maintained in a 12 hr:12 hr light:dark cycle and behavior was monitored for 5 days, of which the last 3 were analyzed for sleep. Pysolo (http://www.pysolo.net) (*Gilestro and Cirelli, 2009*) was used to analyze and plot sleep patterns.

## Statistics

For both behavioral and acylcarnitine metabolomics results, the experimental group was compared to two parental controls by one-way ANOVA with Holm-Sidak post hoc tests (in addition to specific tests mentioned in figure legends) with GraphPad Prism software. For the initial comparisons of metabolomic data, Metabolon performed Welch's t-tests (*Table 1* and *Supplementary file 1*) on scaled signal data for each metabolite, to compare different conditions. Raw signal was scaled so that the median would be equal to 1, using all samples that had been concurrently run. Missing values were filled in with the lowest value of run samples for that metabolite. Additional details of statistics tests are listed in the figure legends. For the qPCR results, the experimental group was compared to control iso31 by Mann-Whitney T test.

## Acknowledgements

We thank Dr. Chris Petucci and the Penn Metabolomics Core for providing measurements of acylcarnitines in fly heads. We thank Zhifeng Yue for her support with the maintenance of fly strains. The work was supported by HHMI and by R01DK120757, as well as by NIH grant R01 NS048471.

## Additional information

### Competing interests

Amita Sehgal: Reviewing editor, eLife. The other authors declare that no competing interests exist.

### Funding

| Funder | Grant reference number | Author |
|---|---|---|
| National Institute of Diabetes and Digestive and Kidney Diseases | R01DK120757 | Amita Sehgal |
| Howard Hughes Medical Institute | | Amita Sehgal |
| National Institutes of Health | R01 NS048471 | Amita Sehgal |

The funders had no role in study design, data collection and interpretation, or the decision to submit the work for publication.

### Author contributions

Fu Li, Conceptualization, Data curation, Formal analysis, Validation, Investigation, Visualization, Methodology, Writing – original draft, Writing – review and editing; Gregory Artiushin, Conceptualization, Data curation, Formal analysis, Validation, Investigation, Visualization, Methodology, Writing – original draft; Amita Sehgal, Conceptualization, Resources, Supervision, Funding acquisition, Project administration, Writing – review and editing

### Author ORCIDs

Fu Li ⓘ http://orcid.org/0000-0002-0879-8008
Gregory Artiushin ⓘ https://orcid.org/0000-0003-1615-3012
Amita Sehgal ⓘ http://orcid.org/0000-0001-7354-9641

### Decision letter and Author response

Decision letter https://doi.org/10.7554/eLife.86336.sa1
Author response https://doi.org/10.7554/eLife.86336.sa2

## Additional files

### Supplementary files

Supplementary file 1. Metabolomics of Repo>20x*Shibire* fly heads. All measured metabolites and their respective categories are listed for samples from *Repo-GAL4*>UAS-20x*Shibire*, and both parental controls. Welch's t-test was performed on scaled signal for each metabolite, comparing the conditions shown. Green highlighting marks a significant difference (p≤0.05) between the groups, where metabolite ratio is <1.00, while light green is not significant, but close to the threshold (0.05<p<0.10). Red highlighting marks a significant difference (p≤0.05) between groups where metabolite ratio is ≥1.00, and light red is not significant, but close to the threshold (0.05<p<0.10).

MDAR checklist

### Data availability

All data generated or analysed during this study are included in the manuscript and supporting file. Source data files have been provided for Figures 1, 2, 3, 4, 5, 6, 7, 8 and Figure 1—figure supplement 1, Figure 2—figure supplement 1, Figure 3—figure supplement 1, 3 ,4, 5 ,6, 7, 8. Figure 4—figure

supplement 1, 2, 3, 4, 5, 6. Metabolomic (mass spec) data reported in the manuscript is available at Zenodo.

The following dataset was generated:

| Author(s) | Year | Dataset title | Dataset URL | Database and Identifier |
|---|---|---|---|---|
| Li F, Artiushin G, Sehgal A | 2023 | Modulation of sleep by trafficking of lipids through the Drosophila blood-brain barrier | https://doi.org/10.5281/zenodo.15086435 | Zenodo, 10.5281/zenodo.15086435 |

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
