## [Editor Report]

Through a convincing series of in vivo experiments, this work demonstrates that lipid- and carnitine transporters in glial cells of the *Drosophila* blood-brain barrier play important roles in the regulation of sleep. The data are consistent with metabolite clearance, in particular, the sleep-dependent transport of acyl-carnitine across the BBB, being an important function of sleep.

---

## [Decision Letter]

**Decision letter after peer review:**

[Editors’ note: the authors submitted for reconsideration following the decision after peer review. What follows is the decision letter after the first round of review.]

Thank you for submitting the paper "Modulation of sleep by trafficking of lipids through the *Drosophila* blood brain barrier" for consideration by *eLife*. Your article has been reviewed by 3 peer reviewers, and the evaluation has been overseen by a Reviewing Editor and a Senior Editor. The reviewers have opted to remain anonymous.

Comments to the Authors:

We are sorry that, after consultation with the reviewers, we have decided that this work cannot be considered further for publication by *eLife*. However, we would be willing to consider an extensively revised manuscript if you choose to address the reviewers' concerns and resubmit to *eLife* as a new submission.

Specifically, as you will see from the comments below, while all of the reviewers found the question and approach of this study to be very interesting, all felt that several additional experiments including routine controls and procedural enhancements as well as careful rewriting would be required. Particularly critical is to attempt experiments to establish causality of acylcarnitine build up and sleep regulation (e.g through acylcarnitine microinjection?). A subset of other issues raised in the reviewer discussions included the need for: (1) more complete description of used methods used; (2) description of all drivers as well as transgenic constructs used, with references as well as validation of driver expression via UAS-GFP if not previously described (primarily to more clearly implicate barrier glia as against other glial cells). (3) validation of RNAi efficacy via qRT (repo>RNAi and head or brain mRNA levels); (4) ideally, key experiments to employ more than one RNAi transgene to ensure that effects are not due to off-target effects, while acknowledging that sometimes these genetics are difficult (e.g. with TRiP lines).

*Reviewer #1 (Recommendations for the authors):*

Artiushin et al. use a combination of metabolomics and behavioral genetics to detail the relationship between sleepiness and lipid transport in *Drosophila*. In previous studies, the Sehgal lab found that disrupting endocytosis in glia of the fly blood-brain barrier is sufficient to increase sleep. Here, the authors sought to explore the mechanism(s) behind this phenomenon by measuring metabolites in the heads of Repo>shi1 flies. LC-MS found elevations in a variety of acylcarnitines in the heads of long-sleeping Repo>shi1 flies, then behavioral experiments found that RNAi-mediated knock-down of two lipid-transport related genes (Lrp1 and Megalin) or two carnitine transporters (Orct and Orct2) are sufficient to significantly increase sleep time. Finally, LC-MS measurements in each long-sleeping RNAi-knock down genotype (Repo>Lrp1RNAi, MegalinRNAi and Repo>OrctRNAi, Orct2RNAi) also showed increased levels of acylcarnitine species. Together, these studies nicely parallel findings of heightened acylcarnitine levels in the blood of sleep-deprived humans, indicating that the fly provides a tractable model for understanding the mechanistic link between fatty acid metabolism and sleep.

While the authors' interpretations are consistent with their findings, the addition of a few key experiments, new analyses of behavioral data, and text edits could strengthen the manuscript.

Main comments:

The authors find sleep effects of manipulating fatty acid transport-related pathways in BBB glia, but it is not clear to me whether the acylcarnitines are moving into the brain or out of it. These possibilities each indicate different (and interesting) possibilities for sleep regulation and/or function. Along similar lines, the authors could test whether acylcarnitine levels are increased in the bodies of sleep deprived flies – this finding might help to distinguish whether these are sleep-promoting signals generated in the periphery or a metabolite that needs to be cleared from the brain.

If the sample material for LC-MS experiments came from whole heads, then it is possible that the head fat bodies contributed to the fatty acid changes shown in Figures 1, 5 & 6. The authors should mention and discuss this possibility in the manuscript.

Do the sleep changes in Figures2-4 alter sleep during the day or night (or both)? Do these RNAis alter sleep architecture or waking locomotor activity?

The authors use Gal80TS to restrict the timing of Mgl/Lrp1 RNAi (Figure 3B-C) and Orct/Orct2 RNAi (Figure 4B-C). A clearer description of the temperature protocols should be included. Do the 18°C and 31°C represent different groups of flies that were tested in parallel? Or do they show the same flies before and after a temperature increase? In addition, the flies housed at 18°C (when RNAi expression should be blocked) show similar behavioral phenotypes as the 31°C condition and as flies expressing the same RNAis without Gal80TS (see panels 3A & 4A, respectively). These data indicate either that the Gal80 repressor is not effectively blocking RNAi expression or that the sleep effects observed here cannot be solely attributed to the RNAi knock-downs.

Does supplementing fly food with acylcarnitines alter sleep? If so, are flies expressing Mgl/Lrp RNAi or Orct/Orct2 RNAi in perineural glia sensitized to dietary acylcarnitines?

Do all error bars indicate SEM as listed in the legends? Several of the behavioral plots appear to show overlapping error bars across comparisons that are designated as statistically significant. See Figures 3B, 3C, 4B, 4C and Figure 3 – —figure supplement 1.

*Reviewer #2 (Recommendations for the authors):*

Here, Artiushin et al. perform metabolomics on the heads transgenic mutants that exhibit long sleep due to inhibition of glial endocytosis (via repo driving shibire) compared to genetic controls. They identified multiple changes in metabolites; the most consistent and significant change was increased acylcarnitine levels. The authors performed a candidate screen by driving RNAi-mediated inhibition of 50 glia-enriched genes with pan-glial or sub-glial expression drivers. Knockdown of lipid transporter proteins in barrier glia led to increased acylcarnitines and increased sleep, suggesting that increased acylcarnitine levels may increase sleep. Previous results had shown that barrier glia and endocytosis were involved in sleep regulation; these data support the hypothesis that lipid metabolites, transported by barrier glia into the brain, mediate sleep regulation. Specifically, they point to the hypothesis that increased acylcarnitine levels in the brain are correlated with induction of sleep.

The conclusions of this paper are mostly well supported but some specific aspects need to be clarified or tested.

1) The n are low for several behavioral experiments and the number of trials is not clear for any experiments. Sleep and behavior are notoriously inconsistent from trial to trial. Experiments should be repeated with 16-32 flies per trial with 3 trials per experiment to demonstrate robustness of results.

2) As negative controls for subglial drivers that drive RNAi expression and affect sleep behavior, the authors should also test a non-barrier sub-glial expression driver such as eye glial drivers for effects on sleep behavior.

3) Are there other methods of increasing acylcarnitine levels in the body or brain that will increase sleep?

4) What are acylcarnitine levels for short-sleeping mutants?

This reviewer appreciates all the data and statistics listed for metabolomics. These data will be invaluable for other researchers in the field. Nevertheless, the precise n and trial number for each experiment should be listed.

*Reviewer #3 (Recommendations for the authors):*

This manuscript is a follow up to a previous publication from the group that linked blood-brain-barrier function to sleep in *Drosophila*. In this new report, they document changes in metabolites that occurs when the fly BBB is disrupted, with acylcarnitines accumulating in particular. Finally, using RNAi, they show that disruption of candidate transporters in the BBB alter sleep.

Overall, the manuscript provides promising data with the potential to be very interesting; however, the current version of this manuscript is hastily assembled. For example, the introduction refers somewhat vaguely to this group's previous paper, of which the current work is an extension, without really giving the reader much context of the state of the field. Although familiar with that previous paper, I had to refer back to it at several points when reading the current document to make full sense of the current manuscript. Particularly poor are the documentation of the methods; an expert in the field would not be able to replicate the findings based on the methods. For just one example-unless I am mistaken (and I misread Flybase), the Shi.ts1 manipulation used to impair the BBB is a temperature sensitive allele, yet there is no mention of samples being placed in restrictive or permissive temperatures. They do an RNAi screen in the middle of this paper, but that screen and its methods are not described. Not even all the fly stocks that appear to be used in the manuscript appear listed in the Methods.

More complete documentation of the experiments being performed would make it easier to properly evaluate the figures and conclusions.

The paper needs a major rewrite of the methods. Other areas that need to be addressed:

In Table 1 (and Figure 1), many of the metabolites labelled as interesting appear to be upregulated to quite a degree when comparing the two control strains, which makes it hard to judge whether these metabolic changes are meaningful or not. They appear to use a t-test for each metabolite, which further confounds the interpretation-some kind of false discovery rate correction should be applied to such large lists of comparisons.

On a related note, the manuscript could do a better job of guiding the reader to how the genetic manipulations are linked to the metabolite profile deemed important/different in the first figures. For example, comparing the results from Figure 5 and 6, it would appear that only C16 and C16.1 are showing differences that are found in Figure 1, but I could only find this upon careful examination myself when the authors should really do a better job to help us make sense of this data. Similarly, Figure 2's results don't seem in any way connected to the metabolite profiles, so I cannot understand why this data is included here.

Other aspects that I found difficult to assess:

– The RNAi screen, were more than one RNAi aimed at the same gene to confirm the effects?

– The raw values for metabolites are labelled as "area under the curve" in Figure 1 (unclear what that means) and plotted as μM in Figure 5.

– The n's for each experiment appear to be 13-16 per genotype. That sounds like a single tracking experiment to me. Were any of these experiments done as independent biological replicates (i.e. repeated on separate days?)

[Editors’ note: further revisions were suggested prior to acceptance, as described below.]

Thank you for resubmitting your work entitled "Modulation of sleep by trafficking of lipids through the *Drosophila* blood brain barrier" for further consideration by *eLife*. Your revised article has been evaluated by K VijayRaghavan (Senior Editor) and a Reviewing Editor.

The reviewers have discussed their reviews with one another, and the Reviewing Editor requests you to consider responding appropriately to the remaining minor queries and comments.

*Reviewer #1 (Recommendations for the authors):*

Specific comments:

1. Does Figure 1 represent data from the same experiments described in Table 1? It's not clear to me whether Figure 1 shows more detail from the same data points or represents additional replicates. Also, the legend for Figure 1 describes the use of T-tests for significance – these should be analyzed using ANOVA instead with post hoc comparisons for the significance of each fatty acid.

2. The authors list the contents of each acylcarnitine type that is contained within the mixtures that are fed to flies, but a clearer discussion in the text that compares these lists with the lipids that were significantly changed in figure 1 might help the reader.

3. Is sleep architecture changed in flies that sleep more with acylcarnitine feeding?

*Reviewer #2 (Recommendations for the authors):*

The authors have addressed all of my concerns from the first version.

*Reviewer #3 (Recommendations for the authors):*

The authors did great work in addressing the majority of prior reviewer comments, specifically by clarifying their Materials and methods and adding supplementary experiments that increase confidence in their findings. The authors find that many different types of gene manipulations in the blood-brain barrier are associated with both long sleep and high acylcarnitine accumulation in the brain; the number of these experiments strongly supports a relationship between these three events. Precisely how these different gene manipulations in the BBB lead to acylcarnitine accumulation in the brain and how/where/why acylcarnitine accumulation in the brain might cause long sleep (as opposed to resulting from long sleep) remain unclear. Thus, while the authors present sufficient and exciting evidence that sleep, BBB function, and acylcarnitines are related, due to the lack of a cohesive model or mechanism, the authors are right to be cautious in making more specific claims about the causal nature of these relationships.

This manuscript was clearer and more well-written than the previous version. This reviewer did not go through with a fine-tooth comb to examine whether claims of causality were overstated; the authors should do this.

I include here a few points that could be mentioned in the discussion, results, or methods sections as caveats or clarifying points:

– Is there any evidence in the literature that acylcarnitines are removed from the brain via the BBB, as would be suggested by acylcarnitine accumulation after the loss of endocytosis or transporter function in BBB? Is this the authors' hypothesis?

– The feeding experiments, though the effects were mild, go one step toward addressing the role of acylcarnitines in triggering sleep except that it's not clear if eating acylcarnitines leads to acylcarnitine accumulation in the brain or if this is a separate effect.

– Another detail that the authors could add: the time of day of collection for metabolomics experiments. The authors might comment in the discussion on how this timing might relate to the mechanism.

---

## [Author Response]

[Editors’ note: the authors resubmitted a revised version of the paper for consideration. What follows is the authors’ response to the first round of review.]

Comments to the Authors:We are sorry that, after consultation with the reviewers, we have decided that this work cannot be considered further for publication by eLife. However, we would be willing to consider an extensively revised manuscript if you choose to address the reviewers' concerns and resubmit to eLife as a new submission.Specifically, as you will see from the comments below, while all of the reviewers found the question and approach of this study to be very interesting, all felt that several additional experiments including routine controls and procedural enhancements as well as careful rewriting would be required. Particularly critical is to attempt experiments to establish causality of acylcarnitine build up and sleep regulation (e.g through acylcarnitine microinjection?). A subset of other issues raised in the reviewer discussions included the need for: (1) more complete description of used methods used; (2) description of all drivers as well as transgenic constructs used, with references as well as validation of driver expression via UAS-GFP if not previously described (primarily to more clearly implicate barrier glia as against other glial cells). (3) validation of RNAi efficacy via qRT (repo>RNAi and head or brain mRNA levels); (4) ideally, key experiments to employ more than one RNAi transgene to ensure that effects are not due to off-target effects, while acknowledging that sometimes these genetics are difficult (e.g. with TRiP lines).

We have included (1) more complete methods; (2) description of drivers, including references for GFP expression of these; (3) validation of RNAi efficacy; (4) use of more than one RNAi transgene wherever possible. We have also conducted other additional experiments recommended by reviewers, as indicated below.

Reviewer #1 (Recommendations for the authors):Artiushin et al. use a combination of metabolomics and behavioral genetics to detail the relationship between sleepiness and lipid transport in *Drosophila*. In previous studies, the Sehgal lab found that disrupting endocytosis in glia of the fly blood-brain barrier is sufficient to increase sleep. Here, the authors sought to explore the mechanism(s) behind this phenomenon by measuring metabolites in the heads of Repo>shi1 flies. LC-MS found elevations in a variety of acylcarnitines in the heads of long-sleeping Repo>shi1 flies, then behavioral experiments found that RNAi-mediated knock-down of two lipid-transport related genes (Lrp1 and Megalin) or two carnitine transporters (Orct and Orct2) are sufficient to significantly increase sleep time. Finally, LC-MS measurements in each long-sleeping RNAi-knock down genotype (Repo>Lrp1RNAi, MegalinRNAi and Repo>OrctRNAi, Orct2RNAi) also showed increased levels of acylcarnitine species. Together, these studies nicely parallel findings of heightened acylcarnitine levels in the blood of sleep-deprived humans, indicating that the fly provides a tractable model for understanding the mechanistic link between fatty acid metabolism and sleep.While the authors' interpretations are consistent with their findings, the addition of a few key experiments, new analyses of behavioral data, and text edits could strengthen the manuscript.Main comments:The authors find sleep effects of manipulating fatty acid transport-related pathways in BBB glia, but it is not clear to me whether the acylcarnitines are moving into the brain or out of it. These possibilities each indicate different (and interesting) possibilities for sleep regulation and/or function. Along similar lines, the authors could test whether acylcarnitine levels are increased in the bodies of sleep deprived flies – this finding might help to distinguish whether these are sleep-promoting signals generated in the periphery or a metabolite that needs to be cleared from the brain.

This is a good idea, thank you. We assume the reviewer meant bodies of RepoGal4>Shi^1^, which were the ones analyzed here, and not sleep-deprived flies. We collected bodies of RepoGal4>Shi^1^, flies and controls and report here that while they show changes in lipid content, the changes are different from those seen in heads. Please see the results in Figure 1- supplement Figure 1.

If the sample material for LC-MS experiments came from whole heads, then it is possible that the head fat bodies contributed to the fatty acid changes shown in Figures 1, 5 & 6. The authors should mention and discuss this possibility in the manuscript.

Good point. We have included discussion of this possibility in our revised manuscript.

Do the sleep changes in Figures2-4 alter sleep during the day or night (or both)? Do these RNAis alter sleep architecture or waking locomotor activity?

Day and night changes in sleep are now shown in Figures2-4. We have also characterized sleep architecture for all behavioral assays and now show this, along with activity index (waking locomotor activity), in Figure 2 and in several supplemental figures for Figures3 and 4. None of the RNAis altered waking locomotor activity.

The authors use Gal80TS to restrict the timing of Mgl/Lrp1 RNAi (Figure 3B-C) and Orct/Orct2 RNAi (Figure 4B-C). A clearer description of the temperature protocols should be included. Do the 18°C and 31°C represent different groups of flies that were tested in parallel? Or do they show the same flies before and after a temperature increase? In addition, the flies housed at 18°C (when RNAi expression should be blocked) show similar behavioral phenotypes as the 31°C condition and as flies expressing the same RNAis without Gal80TS (see panels 3A & 4A, respectively). These data indicate either that the Gal80 repressor is not effectively blocking RNAi expression or that the sleep effects observed here cannot be solely attributed to the RNAi knock-downs.

A description of the temperature protocol has been added to the Methods and figure legend in the revised MS. The same flies were assayed before and after a temperature increase (Figure3 &4). We have done additional repeats of sleep tests and, as shown in Figures 3 and 4, the data indicate that the Gal80 repressor is effectively blocking RNAi expression.

Does supplementing fly food with acylcarnitines alter sleep? If so, are flies expressing Mgl/Lrp RNAi or Orct/Orct2 RNAi in perineural glia sensitized to dietary acylcarnitines?

To address the reviewer’s question, we fed *iso31* flies mixes of acylcarnitines of different chain lengths and monitored sleep behavior. We found that acylcarnitines mixture 1 (C0 and C2) did not alter sleep amount, but low concentrations of acylcarnitine mixtures 2 and 3, which are composed of medium and long chain acylcarnitines, produced a small increase in sleep. Given that sleep is already increased with knockdown of *Mgl/Lrp* or *Orct/Orct2* in perineural glia, we suspect that it would be difficult to detect a further increase with acylcarnitine feeding. Fly sleep easily gets to ceiling levels.

Do all error bars indicate SEM as listed in the legends? Several of the behavioral plots appear to show overlapping error bars across comparisons that are designated as statistically significant. See Figures 3B, 3C, 4B, 4C and Figure 3 – —figure supplement 1.

Sorry about this. Previous figures actually plotted SD in some cases, but we have now fixed this so error bars show SEM, as indicated in the legends of the revised MS.

Reviewer #2 (Recommendations for the authors):Here, Artiushin et al. perform metabolomics on the heads transgenic mutants that exhibit long sleep due to inhibition of glial endocytosis (via repo driving shibire) compared to genetic controls. They identified multiple changes in metabolites; the most consistent and significant change was increased acylcarnitine levels. The authors performed a candidate screen by driving RNAi-mediated inhibition of 50 glia-enriched genes with pan-glial or sub-glial expression drivers. Knockdown of lipid transporter proteins in barrier glia led to increased acylcarnitines and increased sleep, suggesting that increased acylcarnitine levels may increase sleep. Previous results had shown that barrier glia and endocytosis were involved in sleep regulation; these data support the hypothesis that lipid metabolites, transported by barrier glia into the brain, mediate sleep regulation. Specifically, they point to the hypothesis that increased acylcarnitine levels in the brain are correlated with induction of sleep.The conclusions of this paper are mostly well supported but some specific aspects need to be clarified or tested.1) The n are low for several behavioral experiments and the number of trials is not clear for any experiments. Sleep and behavior are notoriously inconsistent from trial to trial. Experiments should be repeated with 16-32 flies per trial with 3 trials per experiment to demonstrate robustness of results.

All the experiments have been repeated at least 3 times and additional numbers have been added.

2) As negative controls for subglial drivers that drive RNAi expression and affect sleep behavior, the authors should also test a non-barrier sub-glial expression driver such as eye glial drivers for effects on sleep behavior.

As suggested by the reviewer, we used non-barrier glial drivers *MZ0709-GAL4, NP2222-GAL4* and *Eaat-GAL4* with tubgal80ts to express *Lrps* or *Orcts* RNAi in ensheathing glia, cortex glia and astrocyte-like glia respectively. None of these affected sleep (Figure 3—figure supplement 5 and Figure 4—figure supplement 2).

3) Are there other methods of increasing acylcarnitine levels in the body or brain that will increase sleep?

We used acylcarnitines as a food supplement for *iso31* and report increases in sleep with some of them (Figure 1—figure supplement Figure 2). We also tried to inject acylcarnitines into fly heads but we could not control the amount and distribution of the injected material.

4) What are acylcarnitine levels for short-sleeping mutants?

We have conducted metabolomic analysis of three short-sleeping mutants *Fmn, Rye* and *SssP1* and found that acylcarnitines are increased in these (Bedont et al., BioRxiv, also under consideration). As we believe these mutants to be chronically sleep-deprived, i.e. they have the need to sleep but are unable to implement it, these data support our hypothesis that acylcarnitines reflect high sleep need.

This reviewer appreciates all the data and statistics listed for metabolomics. These data will be invaluable for other researchers in the field. Nevertheless, the precise n and trial number for each experiment should be listed.

All the experiments have been repeated at least 3 times and new data added.

Reviewer #3 (Recommendations for the authors):This manuscript is a follow up to a previous publication from the group that linked blood-brain-barrier function to sleep in *Drosophila*. In this new report, they document changes in metabolites that occurs when the fly BBB is disrupted, with acylcarnitines accumulating in particular. Finally, using RNAi, they show that disruption of candidate transporters in the BBB alter sleep.Overall, the manuscript provides promising data with the potential to be very interesting; however, the current version of this manuscript is hastily assembled. For example, the introduction refers somewhat vaguely to this group's previous paper, of which the current work is an extension, without really giving the reader much context of the state of the field. Although familiar with that previous paper, I had to refer back to it at several points when reading the current document to make full sense of the current manuscript. Particularly poor are the documentation of the methods; an expert in the field would not be able to replicate the findings based on the methods. For just one example-unless I am mistaken (and I misread Flybase), the Shi.ts1 manipulation used to impair the BBB is a temperature sensitive allele, yet there is no mention of samples being placed in restrictive or permissive temperatures. They do an RNAi screen in the middle of this paper, but that screen and its methods are not described. Not even all the fly stocks that appear to be used in the manuscript appear listed in the Methods.More complete documentation of the experiments being performed would make it easier to properly evaluate the figures and conclusions.

We apologize for the omissions. As noted, we have conducted additional experiments and also extensively re-written the manuscript to address these concerns. In particular, we now indicate that we maintained *shibire* flies for metabolomics at 30℃, even though sleep phenotypes are evident at permissive temperatures also (Artiushin et al., 2018) and we include details of the screen as well as a list of fly stocks in Methods.

The paper needs a major rewrite of the methods. Other areas that need to be addressed:

The methods section has been rewritten in the revised MS.

In Table 1 (and Figure 1), many of the metabolites labelled as interesting appear to be upregulated to quite a degree when comparing the two control strains, which makes it hard to judge whether these metabolic changes are meaningful or not. They appear to use a t-test for each metabolite, which further confounds the interpretation-some kind of false discovery rate correction should be applied to such large lists of comparisons.

Short, medium, long and very long chain acylcarninites are shown in Table 1 and Figure 1. We have used one-way ANOVA with Holm-Sidak post-hoc comparisons to compare these in experimental and control groups in the revised manuscript (Figure 1).

On a related note, the manuscript could do a better job of guiding the reader to how the genetic manipulations are linked to the metabolite profile deemed important/different in the first figures. For example, comparing the results from Figure 5 and 6, it would appear that only C16 and C16.1 are showing differences that are found in Figure 1, but I could only find this upon careful examination myself when the authors should really do a better job to help us make sense of this data. Similarly, Figure 2's results don't seem in any way connected to the metabolite profiles, so I cannot understand why this data is included here.

We apologize for this oversight, and have added comparison of Figure 5-6 data to Figure 1. Regarding Figure 2, we acknowledge that these data are not connected to metabolite profiles, but they were important findings of our transporter screen and we feel they should be reported. If the reviewer feels these data should be in the supplement, we can move them.

Other aspects that I found difficult to assess:– The RNAi screen, were more than one RNAi aimed at the same gene to confirm the effects?

The revised manuscript reports the use of multiple RNAis for Lrp1/Mgl and Orct/Orct2. We have now also checked the knockdown efficiency of RNAi lines by qPCR (Figure 3—figure supplement 2 and Figure 4—figure supplement 3).

– The raw values for metabolites are labelled as "area under the curve" in Figure 1 (unclear what that means) and plotted as μM in Figure 5.

The metabolite data were from two different facilities (Metabolon and The Penn Metabolomics Core). They used different methods, ’area under the curve’ by Metabolon and μM by the Penn Metabolomics Core to show raw values.

– The n's for each experiment appear to be 13-16 per genotype. That sounds like a single tracking experiment to me. Were any of these experiments done as independent biological replicates (i.e. repeated on separate days?)

We show additional data in the revised MS. The sleep data are from at least 3 independent trials and about 48 flies per genotype.

[Editors’ note: what follows is the authors’ response to the second round of review.]

Reviewer #1 (Recommendations for the authors):Specific comments:1. Does Figure 1 represent data from the same experiments described in Table 1? It's not clear to me whether Figure 1 shows more detail from the same data points or represents additional replicates. Also, the legend for Figure 1 describes the use of T-tests for significance – these should be analyzed using ANOVA instead with post hoc comparisons for the significance of each fatty acid.

Yes. Figure 1 represents data from the same experiments described in Table 1. We did indeed use one-way ANOVA with Holm-Sidak post-hoc comparisons, and not T-tests, for significance in figure 1. Table 1 compares the experimental to each of the parental controls separately and so uses a T-test.

2. The authors list the contents of each acylcarnitine type that is contained within the mixtures that are fed to flies, but a clearer discussion in the text that compares these lists with the lipids that were significantly changed in figure 1 might help the reader.

We have added a comparison of acylcarnitine mixtures (from Σ) and the ones that are significantly changed in figure1.

3. Is sleep architecture changed in flies that sleep more with acylcarnitine feeding?

We have added the analysis of sleep architecture in flies fed acylcarnitines, in multiple additional supplemental figures.

Reviewer #3 (Recommendations for the authors):The authors did great work in addressing the majority of prior reviewer comments, specifically by clarifying their Materials and methods and adding supplementary experiments that increase confidence in their findings. The authors find that many different types of gene manipulations in the blood-brain barrier are associated with both long sleep and high acylcarnitine accumulation in the brain; the number of these experiments strongly supports a relationship between these three events. Precisely how these different gene manipulations in the BBB lead to acylcarnitine accumulation in the brain and how/where/why acylcarnitine accumulation in the brain might cause long sleep (as opposed to resulting from long sleep) remain unclear. Thus, while the authors present sufficient and exciting evidence that sleep, BBB function, and acylcarnitines are related, due to the lack of a cohesive model or mechanism, the authors are right to be cautious in making more specific claims about the causal nature of these relationships.This manuscript was clearer and more well-written than the previous version. This reviewer did not go through with a fine-tooth comb to examine whether claims of causality were overstated; the authors should do this.I include here a few points that could be mentioned in the discussion, results, or methods sections as caveats or clarifying points:– Is there any evidence in the literature that acylcarnitines are removed from the brain via the BBB, as would be suggested by acylcarnitine accumulation after the loss of endocytosis or transporter function in BBB? Is this the authors' hypothesis?

This is an interesting point. We were not able to find anything in the literature that says acylcarnitines are removed from the brain via the BBB. It is possible that this uptake is specific for glial cells (*Drosophila* BBB is composed entirely of glia). Please note too that although the role of the BBB is supported by the sleep phenotypes we get with knockdown of lipid/carnitine transporters specifically in the BBB, the metabolomic analysis was conducted with manipulations of all glia.

– The feeding experiments, though the effects were mild, go one step toward addressing the role of acylcarnitines in triggering sleep except that it's not clear if eating acylcarnitines leads to acylcarnitine accumulation in the brain or if this is a separate effect.

It is true that at this point we do not know if the fed acylcarnitines accumulate in the brain. We have acknowledged this point in the manuscript.

– Another detail that the authors could add: the time of day of collection for metabolomics experiments. The authors might comment in the discussion on how this timing might relate to the mechanism.

The flies were collected at ZT 0-3 for the metabolomics analysis. We have added discussion of how the timing relates to the results.